# Deep learning shows declining groundwater levels in Germany until 2100 due to climate change

Andreas Wunsch [1✉], Tanja Liesch [1] & Stefan Broda [2]

In this study we investigate how climate change will directly influence the groundwater resources in Germany during the 21st century. We apply a machine learning groundwater level prediction approach based on convolutional neural networks to 118 sites well distributed over Germany to assess the groundwater level development under different RCP scenarios (2.6, 4.5, 8.5). We consider only direct meteorological inputs, while highly uncertain anthropogenic factors such as groundwater extractions are excluded. While less pronounced and fewer significant trends can be found under RCP2.6 and RCP4.5, we detect significantly declining trends of groundwater levels for most of the sites under RCP8.5, revealing a spatial pattern of stronger decreases, especially in the northern and eastern part of Germany, emphasizing already existing decreasing trends in these regions. We can further show an increased variability and longer periods of low groundwater levels during the annual cycle towards the end of the century.

[1] Karlsruhe Institute of Technology, Karlsruhe, Germany. [2] Federal Institute for Geosciences and Natural Resources, Berlin, Germany. ✉email: andreas.wunsch@kit.edu

The climate crisis is increasingly altering water availability even in generally water-rich areas like Germany, where overall water stress is currently low[1]. Nevertheless, hot and dry summers in recent years (especially 2018–2020) led to ongoing exceptional droughts[2,3] with severe consequences for agriculture and ecology, such as drought damages in forests, reduced crop yields, and extreme low flows in rivers. Drought effects accumulated over the years, because winter precipitation did not compensate for summer deficits. This applies not only but especially to groundwater resources, which constitute the major source of drinking water supply in Germany (almost 70%)[4]. Declining groundwater levels due to generally reduced groundwater recharge and higher water demand in summer regionally forced water suppliers to exploit their current maximum capacity during dry periods to meet the demand; locally, even water supply shortages occurred. During future dry periods, strong usage conflicts can be expected in areas of low water availability between water suppliers and industry (process and cooling water), additionally amplified by increasing agricultural irrigation demand, which currently has only minor significance with less than 2% of the total withdrawal volume[1]. Knowledge of future groundwater level development, especially in the long-term, is, therefore, crucial to develop sustainable groundwater management plans to meet future demands, solve usage conflicts and protect ecosystems.

Climate change affects groundwater in several direct and indirect ways[5]. Major direct drivers are changes in precipitation, snowmelt, and evapotranspiration[6]. Different representative concentration pathways (RCP) describe possible future climate scenarios. The current situation best matches RCP8.5, often described as a business-as-usual scenario with increasing greenhouse gas emissions. Despite existing mitigation efforts, this scenario might be the most plausible one for the near future[7]. RCP 2.6, a stringent mitigation scenario with an average global warming below 2 °C above pre-industrial temperatures, might be hard to reach at all, and even the intermediate RCP4.5 is still more ambitious than current (as of 2021) nationally determined contributions under the Paris Agreement, according to UN-FCCC[8]. Their analyses estimate global warming of approximately 2.7 °C compared to pre-industrial temperatures. For Germany, analyses based on climate projections show opposing trends in terms of water availability. With some differences between drier and wetter models they find a slight increase in annual precipitation sums, i.e., generally more water, but at the same time with high agreement between models a significant temperature increase of several degrees Celsius by 2100[9–11], i.e., less water. The resulting effect on groundwater resources is therefore not directly clear and needs to be analyzed. Higher precipitation is generally expected during winter, which in combination with a decreasing amount of snow, thus increasing direct infiltration, leads to higher groundwater recharge during winter and less in spring. For the few snow-dominated regions in Germany (e.g., in the South) this might cause changes in seasonality[6], however, overall this plays a minor role. Weather extremes are expected to intensify; therefore, longer droughts and more frequent intense rainfall events will occur[5]. Generally, higher temperatures cause higher atmospheric water demand, thus increasing evapotranspiration, which leads to less infiltration and, therefore, less groundwater recharge. Especially unconfined, shallow aquifers are most likely to be sensitive to direct climate change effects[12]. Indirect climate change influences on groundwater are mostly related to anthropogenic groundwater withdrawals or associated with land-use changes[5]. It is known that the groundwater storage reduction caused by pumping could easily far exceed natural recharge[6,13]. The impact of these factors will be exacerbated as water demand increases to as well meet the needs of a regionally growing population (mainly due to growing urban areas), as of industry and agricultural irrigation. To date, there are no reliable data available that estimate the future development of such factors under different climate change scenarios.

In recent years, artificial neural network (ANN) approaches have proven their usefulness in predicting groundwater levels[14–19], even using a highly transferable approach with purely climatic input variables (e.g., ref. [14]). In a previous study[14], we showed that 1D-convolutional neural networks (CNNs) are a good choice for groundwater level simulation because they mostly outperform even long short-term memory (LSTM) models in terms of accuracy and calculation speed, as well as they showed considerably higher flexibility and modeling stability compared to NARX models (nonlinear autoregressive models with exogenous inputs). Therefore, they are therefore an accurate, fast, and reliable method of choice for this study. Unlike physically-based models, which usually require a very good knowledge of local conditions and need to be time-consumingly built and calibrated, data-driven models such as ANNs can predict a target variable using only relevant driving forces. This makes studies on larger areas easier and is, therefore, the favored approach for this study. To the authors' knowledge, no comprehensive direct evaluation of groundwater level development until 2100 exists for Germany yet. Besides a rather old small-scale study[20] also a regional-scale study for the Danube basin has been conducted to date[21]. The latter uses several dynamically coupled, process-based model components and the authors found strongly declining groundwater levels with declines of up to 10 m close to the Alps in southernmost Germany for their scenario period (2036–2060). Further, several studies investigated future groundwater recharge in different contexts for smaller subregions of Germany using mainly water balance models or process-based models[21–26]. The application of ANNs to study groundwater level development in the long-term and in the context of climate change for a larger area like Germany has not been performed yet. Related studies with applications of ANNs either used a very small number of wells[27–29] and limited time horizons[27,28] or use ANNs without directly presenting future climate signals to the ANN[29]. In the case of streamflow runoff simulation, however, ANNs have been successfully applied to analyze the future development under climate change influences in several catchments all over California[30] as well as in two catchments in China[31,32].

In this study, we use a 1D-CNN approach[14] to build 118 site-specific models, well distributed over Germany in the respective uppermost unconfined aquifer, which are able to predict weekly groundwater levels with high accuracy using only precipitation and temperature as inputs in the past. We visually check the model output plausibility under an artificial extreme climate scenario in the past[30] and investigate how the model has learned input–output relationships using an explainable AI approach (SHAP[33]). We then use the trained CNN models to investigate the future climate-driven groundwater level development for the selected sites, using precipitation and temperature derived from different RCP scenarios (2.6, 4.5, 8.5)[34] of bias-corrected and downscaled ($5 \times 5$ km²) climate projection data[35] from different climate models. These models ("core-ensemble") were preselected by the German Meteorological Service (DWD) to represent 80–90% of the spread of the full ensemble of all available and suitable (according to certain quality criteria) climate projection results under the respective RCP scenario for Germany[36] based on CORDEX-EUR11[37] and ReKliEs-De[38] (see Methods section). As we use purely climatic input variables we can only project the influence of direct climate change effects, while secondary, most certainly stronger indirect effects, such as increased groundwater pumping, are not included in this study. However, due to high prediction accuracy in the past, the selected sites show a strong

relationship between climate variables and groundwater and are unlikely to be under the influence of strong groundwater withdrawals or comparable effects. They are, therefore, suitable for predicting that part of the future groundwater level trend that results from direct climatic influences, as long as the basic input–output relationships remain unchanged.

## Results

**Individual projection results**. For each of the examined 118 test sites, we simulated the future weekly groundwater level development based on 5–6 climate projections per RCP scenario. Since these climate projections differ considerably in detail for individual future time periods, we also obtained several different future groundwater level simulations per scenario and considered site, which should only be interpreted based on longer time periods (at least 30 years)[39], such as with a linear trend analysis performed here, considering the whole time period of more than 80 years. Figure 1 depicts the results of our analysis for RCP8.5, in terms of the relative change in % between the start (2014) and the end of the simulation period (2100) for each of the six projections under RCP8.5 for (a) the annual mean, (b) the annual upper extreme (97.5%) quantile, and (c) the annual lower extreme (2.5%) quantile. For each site, all displayed developments are ordered by the strength of the change, which does not necessarily correspond to the numbering of the projections (Table 1). The given boxplots in Fig. 1d provide more detailed information on the three maps, as well as confidence intervals on the statistical analysis. The values of the non-significant trends are not shown in the boxplots, which has to be kept in mind for interpretation. For detailed numbers on the boxplots, we refer to Supplementary Table S3.

In the case of the annual mean, approximately 47% of all simulations (332 of 708, i.e., six projections for each site) show a significant trend until 2100. There is always at least one result for each site significant ($p < 0.05$), which, however, also means that there are several sites with mainly non-significfant trends (gray). The large majority of the significant trends are negative with a median ranging between −18% (p1) and −6% (p6). Note that the uncertainty (shown by the boxplots in Fig. 1d) can be quite high from the trend analysis alone and we further see that the lower bound sometimes shows a larger spread, thus a higher uncertainty, than the upper bound. In Fig. 1d we also observe that p1 systematically shows the strongest declines until 2100, which is significant for 114 of the 118 wells. The overall maximum decline of the annual mean is -35%, clearly indicating the different character of p1 compared to the other projections. Especially p3–p5 show more moderate changes of the mean (median ranges from −8% to −11%), with many non-significant trends (50–58%). Simulations based on p2 and p6 only find significant trends for 22% and 29% of all sites respectively and additionally are moderate in their significant results. Three projections (p2, p3, but mainly p6) even show some positive trends until 2100, however, overall, these are rare and occur at sites, where other projections simultaneously show at least non-significant or even negative trends. In absolute numbers, the median changes are in the order of −0.1 m to −0.3 m, which is highly dependent on the individual groundwater level range at each site. Despite many non-significant and some positive trends, there is a clear tendency of declining mean groundwater levels until 2100. Additionally, we can observe a slight spatial tendency with more and stronger significant negative trends in some areas of northern and eastern Germany, where we also find the strongest overall relative declines. In southern Germany, many wells show multiple non-significant trends, and most of the positive changes are also scattered in this region; however, some

of the southernmost wells also show some very strong decreases for single simulations.

The results for the upper extreme value quantile (97.5%) confirm these spatial patterns partly. In Fig. 1b we clearly observe many significant declines in eastern Germany, while the large majority (76%) of the trends in whole Germany is considered to be non-significant. Increasing trends are found comparably often, with increases close to 18% (p1, p3, p6). Comparing the projections (Fig. 1d), we find a similar behavior as before: p1 shows the strongest significant decreases (down to −40%, conf. -interval: −61% to −19%), p3–p5 tend to move in the moderate negative range (medians around −11%), while p2 and p6 more often show positive trends (positive medians of the significant trends). Particularly the latter causes a partly contradictory development of the upper extreme values compared to the mean. The absolute numbers of changes are again in the order of a few tens of centimeters.

The tendency of declining groundwater levels we observed for the mean, gets clearer for the lower extreme values (2.5% quantile) shown in Fig. 1c. We still observe 38% non-significant trends, however the remaining 62% show almost exclusively negative changes with a maximum decline of -79%. The median change of the 2.5% quantile of all projections ranges between −34% for p1, which again shows the strongest declines, followed by p4 (−19%), as well as p2, p3, p5, and p6 with a median change around −9% to −12% each (lower bound: −20%, upper bound: −2%). The latter four, and especially of them p6, contain the majority of non-significant trends, the changes shown in the boxplots, therefore, tend to be overestimated. There are only a few sites where only one result is considered significant. These occur, for example, near the Baltic Sea coast as well as the central and eastern part of northern Germany. Quite strong relative decreases are visible in eastern Germany and in the western part of northern Germany as well as at the southernmost sites. This pattern is largely consistent with the spatial pattern of the mean mentioned above. When translating into absolute units, most median decreases (p2–p6) are in the order of −0.1 m to −0.4 m. For p1 and when considering the annual lower extreme value quantile, the median decrease reaches even −0.6 m. From all projections except p6 we see that of all significant changes for the 2.5% quantile, at least a decrease of −0.1 m is observed (summarized in Supplementary Table S3).

The spatial patterns in Fig. 1a–c are particularly interesting because they do not intuitively follow from the patterns of the input data (compare Figs. S7 and S8). Considering all results of RCP8.5, we see a clear tendency toward declining groundwater levels overall, with stronger declines for lower quantiles, i.e., groundwater level lows will occur more frequently and will be more severe in the future. At the same time, except for East Germany, mostly no or even increasing trends are found for upper extreme values, which means that the overall variability will increase considerably by the end of the century.

Figure 2 summarizes the results for the other considered RCP scenarios 2.6 and 4.5. For the former, which is a stringent mitigation scenario in terms of greenhouse gas emissions, we see that generally, the number of significant samples ($p < 0.05$) in total is low, with only 6–8%, depending on the quantile considered. We generally see smaller decreases compared to RCP8.5; the upper extreme value quantile does no longer show considerable positive changes. Supplementary Fig. S1 shows the spatial distribution of the found changes. We can detect no spatial pattern for the 2.5% quantile, but (slight) decreases all over Germany, dominated by mostly non-significant results. The mean and the 97.5% quantile, however, show that decreasing changes occur preferably in northern Germany, whereas the southern part either shows few slight decreases for the mean or remains mostly

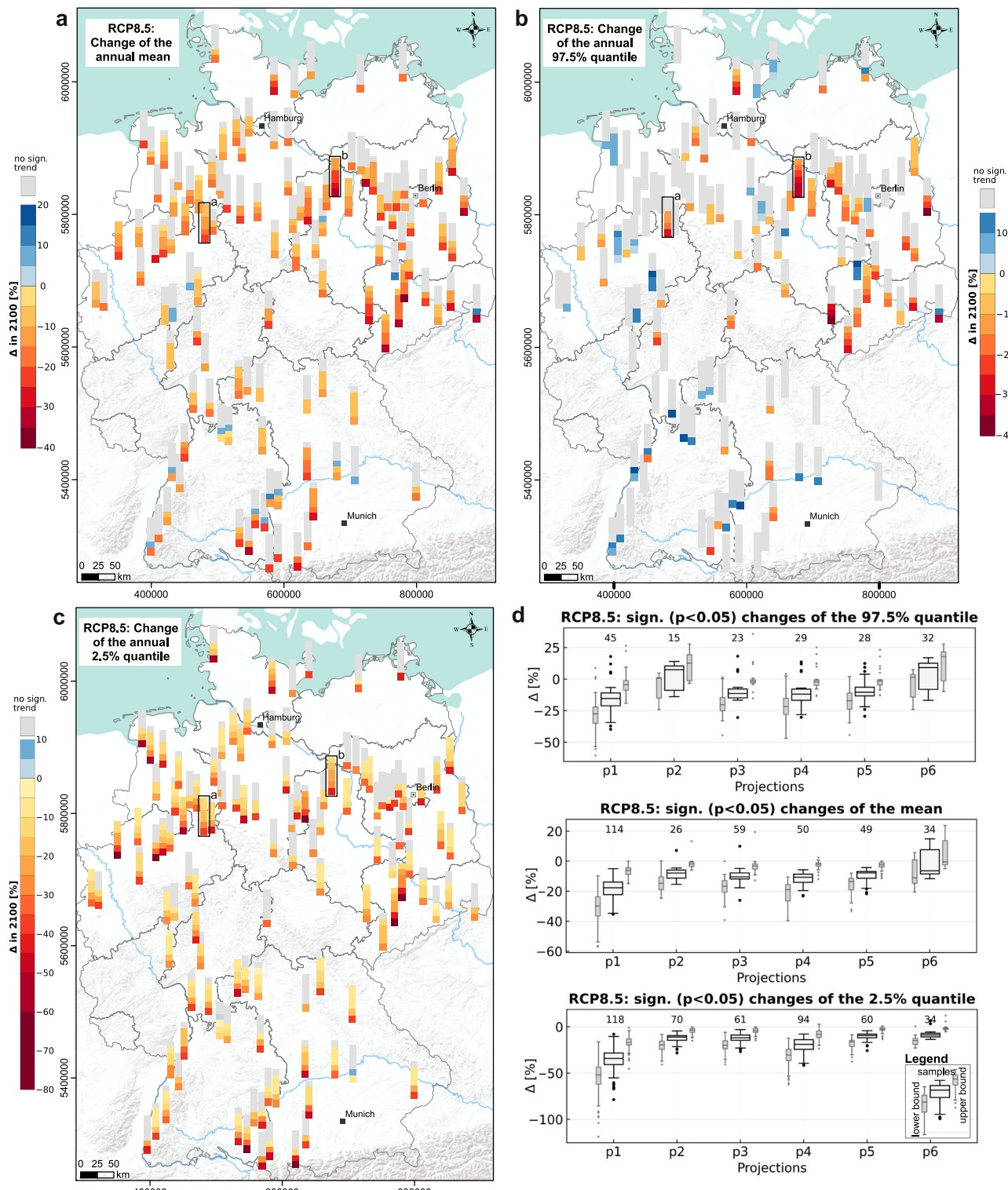

**Fig. 1 Groundwater level changes (RCP8.5).** Change of groundwater levels [%] in 2100 relative to 2014 (start of sim.) for each site and each climate projection, based on linear trend analysis: **a** mean, **b** 97.5% quantile, **c** 2.5% quantile; the order corresponds to the strength and sign of the change. **d** Boxplots showing the significant changes for **a**–**c**, light gray/sideways boxplots show the uncertainty of the change as 95% confidence interval. The numbers above boxplots depict the sample size (significant trends). Black boxes on maps depict the sites shown in Fig. 3.

non-significant for the upper extreme values. The results strongly indicate that the reduced greenhouse gas emissions of the RCP2.6 scenario also translate to a distinctly reduced impact on the groundwater level development, especially compared to the opposite RCP8.5 scenario. Nevertheless, decreasing trends are still visible all over Germany, showing that even for RCP2.6 with limited global warming below 2 °C compared to pre-industrial temperatures, a change in water availability is to be expected.

For RCP4.5 changes are also only rarely significant (Q97.5: 6%, mean: 7%, Q2.5: 13% of all samples). Projection p6 represents

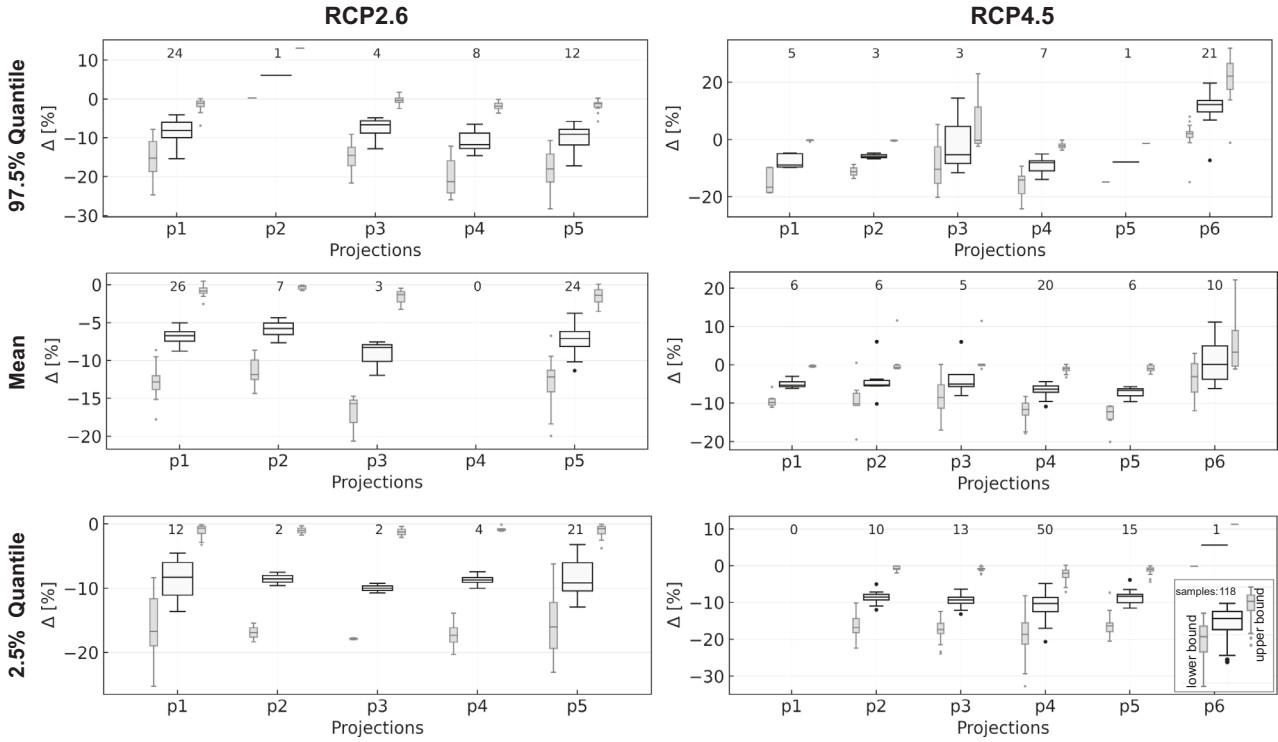

**Fig. 2 Groundwater level changes (RCP2.6, RCP4.5).** Boxplots showing the significant ($p < 0.05$) relative changes between 2014 and 2100 based on linear trend analyses of annual quantiles (2.5% and 97.5%) and the annual mean under RCP2.6 (left) and RCP4.5 (right). Light gray/sideways boxplots show the uncertainty of the change as a 95% confidence interval. The numbers above boxplots depict the sample size (number of significant trends).

definitely an increasing groundwater scenario for the future, whereas p1–p5 mostly show decreases for the significant changes. Except for p6, we, therefore, see median changes of all three annual quantiles between −5% and −10%. RCP4.5 and RCP2.6 do not differ here very strongly, but the number of significant samples is a bit higher for RCP4.5 as well as the confidence intervals are shown in Fig. 2 are slightly narrower than in RCP2.6. Differences get clearer spatially, where we find more distinct patterns in the case of RCP4.5 (Supplementary Fig. S2) with increasing values almost exclusively in southern Germany (97.5% quantile, less frequent also for the mean). This clearly coincides with the spatial pattern of increasing precipitation in the input data (Supplementary Figs. S5 and S6). While decreasing changes can be found in northern Germany for the 2.5% quantile, this is less pronounced for the annual mean and even lesser for the 97.5% quantile. For both, sites with exclusively non-significant changes increasingly dominate. For both, RCP2.6 and 4.5, we do not find the strong decreasing trends in eastern Germany seen for RCP8.5, however, both scenarios indicate a stronger tendency of decreasing trends in the North, a slightly increasing tendency of upper extreme values for the South, as well as an increasing overall variability (decreasing lower quantiles, constant or increasing upper quantiles) are possible. While for RCP2.6 we do not see that the lower extreme values decrease stronger than other parts of the hydrographs as under RCP8.5, this pattern emerges under RCP4.5 in agreement. Overall, due to the high number of non-significant results, RCP2.6 and RCP4.5 results should be interpreted carefully. Maps, as well as detailed numbers on the boxplots in Fig. 2, are part of the electronic supplement (Figs. S1 and S2, Tables S4 and S5).

Figure 3 shows exemplarily the detailed development at two arbitrarily selected sites (black boxes in Fig. 1) under RCPs 4.5 and 8.5, which, as explained, are the most relevant given the current situation. The simulation results are depicted as time series plots for the far future (2071–2100), and as heatmaps with years as rows and weeks as columns for each of the projections. Heatmaps of both scenarios share the same color scale per site. Heatmaps and time series plots of the simulation results of all other sites and for all RCPs are available online (ref. [40]). The time series plots show the diverging development of some projections in the far future, however, there is no strict sequence of projections in terms of absolute groundwater height, the order can change throughout the years. Most heatmaps visualize the development described above by displaying generally declining groundwater levels (more and darker red, as well as lighter or constant blue shadings toward 2100 in the lower part of the heatmaps). Moreover, we observe increasing lengths of periods with low groundwater levels (wider red shadings) throughout the year. In accordance, wet periods usually get shorter (narrower blue shadings) or even change to red (e.g., in b, RCP8.5, p1, p3, p4). The absolute height of groundwater levels during wet periods does not necessarily decrease but can even show the opposite behavior (darker blue, e.g., in a, RCP8.5, p6). Most importantly, in both scenarios and at both sites, we can also recognize successions of several dry years. Such periods are visible in the time series plots, but more clearly as dark red horizontal stripes in the heat maps. These are especially critical because drought effects accumulate and dependent ecosystems cannot recover but are instead particularly vulnerable to further damage in subsequent years due to reduced resilience. Although the results should not be interpreted over shorter periods of time (i.e., they do not reflect the absolute timing of an event), they definitely show the increasing probability of such longer-term droughts in the future, especially in the second half of the century.

**Average projection results under RCP8.5.** In Fig. 4, we consolidated the separate projection results under RCP8.5 for each

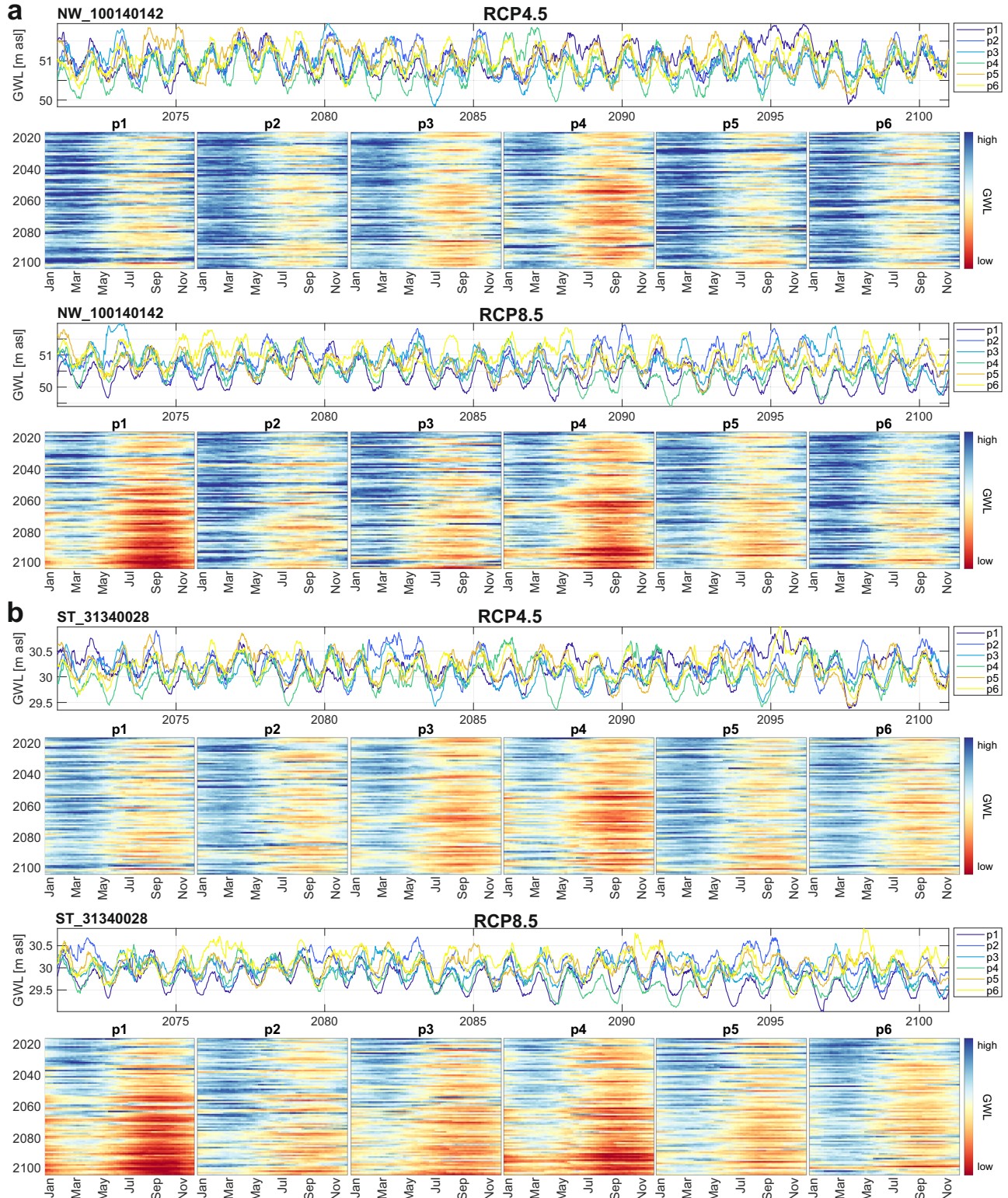

**Fig. 3 Heatmap plots and far future groundwater levels.** RCP4.5 and RCP8.5 results for two arbitrarily selected sites marked by black boxes in Fig. 1 (a NW_100140142, b ST_31340028). Heatmap plots show the whole simulation period for each of the projections under each of the considered scenarios. Columns of each plot as weeks during the year and rows as the year (top: 2014—bottom: 2100).

site into one, by calculating the mean of the significant trends shown in Fig. 1. Only sites with at least four (thus the majority) significant projection results are included, the rest is depicted as not significant on average. This is one reason for neglecting RCPs 2.6 and 4.5 in this analysis step, as barely sites with four or more

significant results were found there. Another reason is that, at least for the near-future, the results of RCP8.5 can be considered most relevant, as it is the scenario closest to our current situation[7]. Even though we investigate a longer time period until 2100, tendencies should be nevertheless useful to estimate near-

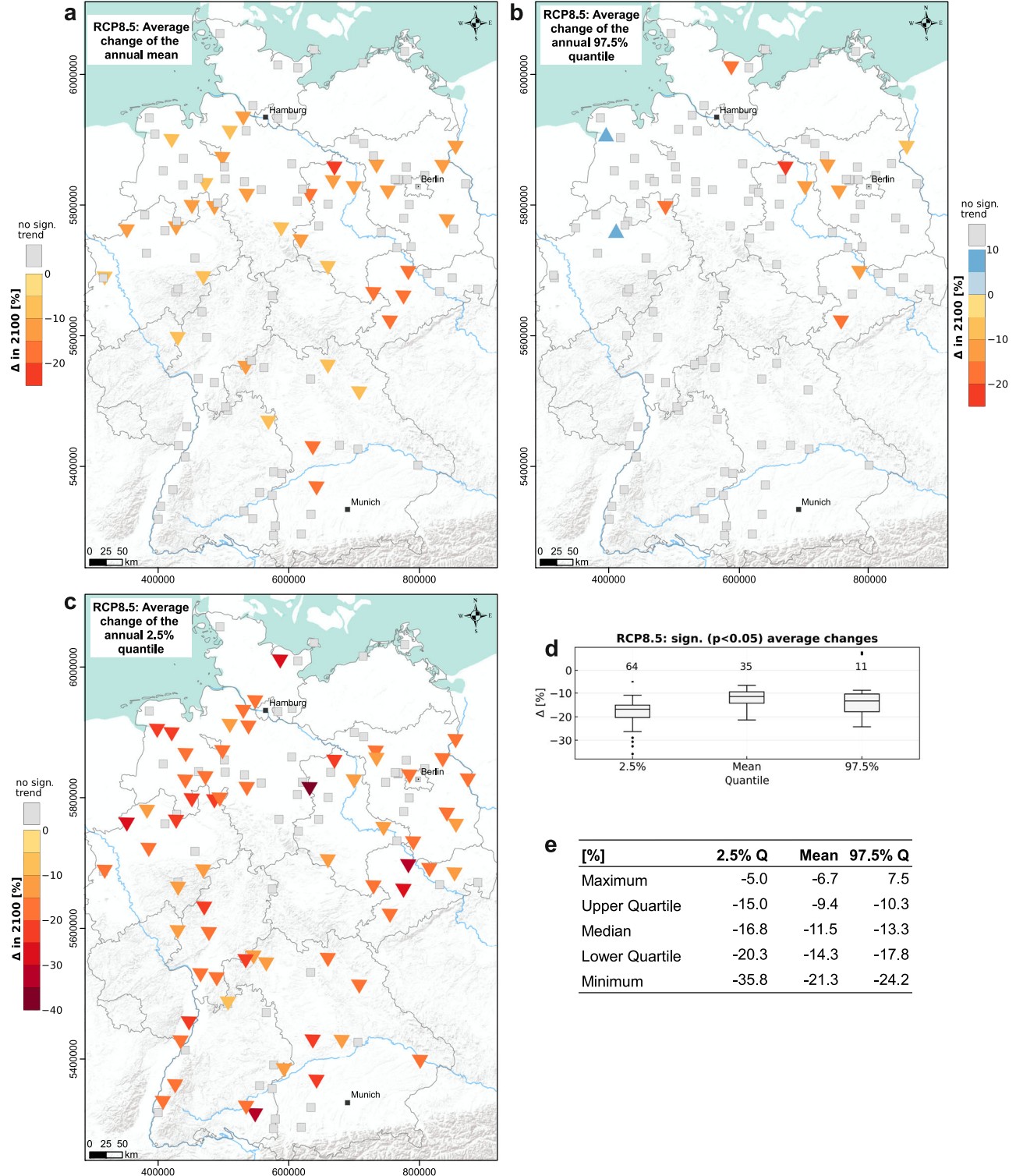

**Fig. 4 Average Changes (RCP8.5).** Averages for all sites of the significant trends (at least four) of the **a** annual mean, **b** the annual 97.5%, and **c** the annual 2.5% quantiles shown also in Fig. 1. **d** Associated boxplots and **e** the corresponding table.

future developments. The development of the mean is depicted in the upper left map (a) and according to the aforementioned definition, about 30% of the wells (35 of 118) are considered significant on average, and on median show a change of −12%. We do not find any wells with increasing mean trends on average and observe a similar spatial pattern as before with the strongest decreases in eastern Germany. For wells in southwestern Germany, we observe a noticeable number of non-significant

changes. Overall, we simulated significant absolute average decreases between −0.2 m and −2.1 m for about 18 wells, and at least a decrease of −9 cm for all 35 wells in Fig. 4a. In the case of the annual 97.5% quantile, the consolidated results show mainly no trends, especially for southern Germany. Two sites in northern Germany are expected to show increased upper extreme values up to a maximum of 7.5% or 0.2 m, however, we still observe a spatial pattern of decreasing upper extreme values in eastern

Germany up to −24%. Hence, in this area, the groundwater levels possibly decrease in every part of the annual cycle and with comparably high certainty (many consistent significant simulations). This also applies to the lower extreme values (2.5% quantile) that show on average significant decreases for more than half of the examined sites, with median decreases of −17% (or −0.3 m) (compare Fig. 4d, e). On this map, no clear spatial pattern is recognizable any longer.

**Model input analysis.** From the combined analysis of our groundwater level simulations, especially under RCP8.5, and the model inputs presented in the data section and Supplementary Figs. S3–S8, we conclude that for shallow aquifers temperature is the main driving factor for declining groundwater levels, rather than precipitation. This applies because mostly non-significantly changing or even increased precipitation is projected, however, our models still frequently show declining groundwater level tendencies. Therefore, these are most likely caused by the significantly increased temperature until the end of the century. Nevertheless, especially under RCP4.5, spatial precipitation data patterns from the input data translate into related patterns of groundwater levels, which shows the also high importance of precipitation. Our results are consistent with other studies, which indicate that the reduction in water availability in the future is driven primarily by changes in temperature[9]. This is also reflected in the results of the model interpretability approach (SHAP[33] values) that we used to check the plausibility of our model outputs. The minimum SHAP value for T is mostly lower than the minimum SHAP value observed for P (except for eight sites); i.e., the models have learned that high temperatures can cause stronger decreasing groundwater levels than low precipitation. This is, however, only an interpretation of what was learned, which agrees with our conception. Causality cannot be derived from this.

**Sources of uncertainty.** There are different sources of uncertainty in our study. Considering the groundwater model itself there exists uncertainty directly from different model realizations as well as the uncertainty due to limited model precision. The former was derived from a Monte-Carlo dropout approach and is on average consistently small for all models (orange sections Fig. 8a and Supplementary Figs. S9–S126), the latter is hard to generalize, as it is different for each site. However, we only used models with high performance in the past, checked the conceptual correctness of what was learned using SHAP values, and investigated the stability of the model output in the extrapolating regime, to improve the confidence in the model simulations. However, it is important to mention that data-driven models generally have difficulties in predicting extreme values. Figure 5 shows the yearly relative model bias on different quantiles during the model testing period (2012–2015, normalized on the historic min–max range of each individual time series). On average the models show a very small bias, however, a considerable bias occurs for extreme values (2.5% and 97.5% quantiles). Lower extremes are overestimated by 4.8%, upper extremes are underestimated by 9.6% (both on median). Thus, the analyses of future extreme values are less robust than for the mean. Nevertheless, we argue that (i) reasonable conclusions can still be derived from relative trends and tendencies even for the extreme values at each site and (ii) since the extreme values are underestimated, the analyses constitute a kind of best-case scenario.

Concerning the simulation of climate change impact, we are not extrapolating in a classical sense, because mean values and frequencies of input values change in the future, but the total range of these values is usually already present in the training

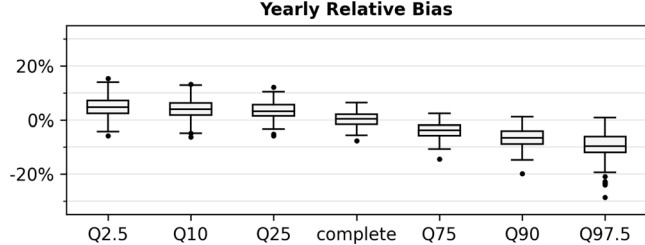

**Fig. 5 Model Bias.** Evaluation of the yearly, relative model bias on different yearly quantiles at all sites for the 4-year model testing period.

data. Scaling uncertainty due to the differences between a single location and the grid cell sizes are certainly present, however, by achieving high performance in the past using training data in the same grid resolution we can assume that this influence is not severe. To account for atmospheric process scales in the climate models that are not reliably scaling down to cell resolution, we follow the DWD best practice recommendation of considering 3 × 3 cells rather than one cell that best matches the site location. Regarding the uncertainty deriving from climate models or the considered scenario themselves, we consider different RCP scenarios each based on a whole ensemble of individual climate models. Finally, the uncertainty from the applied statistical tests (Mann–Kendall test and Theil–Sen slopes) is directly communicated in the text and figures.

## Discussion

The results of our simulations show a nationwide decrease in climate-driven groundwater levels by the end of the century under the RCP8.5 scenario. The results for RCP2.6 and RCP4.5 show comparably few significant changes, thus having to be interpreted with care in absolute and relative numbers. However, this also means that mitigation of greenhouse gas emissions could have a visible effect, at least for the climate-driven part of the total future groundwater levels in Germany. Nevertheless, even for RCP2.6, decreases in all considered quantiles were found all over Germany for some projections. We, therefore, will probably have to cope with drought effects and changing water availability in any of the investigated scenarios, especially because current estimations of future climate change impacts[8] still exceed the RCP4.5 scenario. Especially for the near future, the results under RCP8.5 are most relevant[7], because its path is closest to our current situation.

The absolute changes even under RCP8.5 may seem small, but the facts that we investigated almost exclusively shallow aquifers and sites with comparably small depths to groundwater reinforces the importance of the results, predominantly in terms of water availability for vegetation and agriculture. A decline of several tens of centimeters (depending on the projection and the area) can be vital for plants during hot and dry periods, if, as a result, the groundwater is no longer accessible. Furthermore, a related study showed, that for large parts of northern Germany, a decline of the groundwater levels by 10 cm can be considered critical in terms of altered streamflow discharge due to reduced baseflow from groundwater[13]. This has already been visible during the summers of 2018–2020, when simultaneously to low groundwater levels, also extremely low water levels in surface waters were observed[3]. Our results show a clearer tendency of declining groundwater levels in the North and the East compared to the South (Fig. 4a), which emphasizes the already existing trends and patterns. However, in the southernmost part of Germany, for some individual projections, we also find some of the strongest declines (Fig. 1). It is very important to note that the assessed results are only long-term averages of future development. As

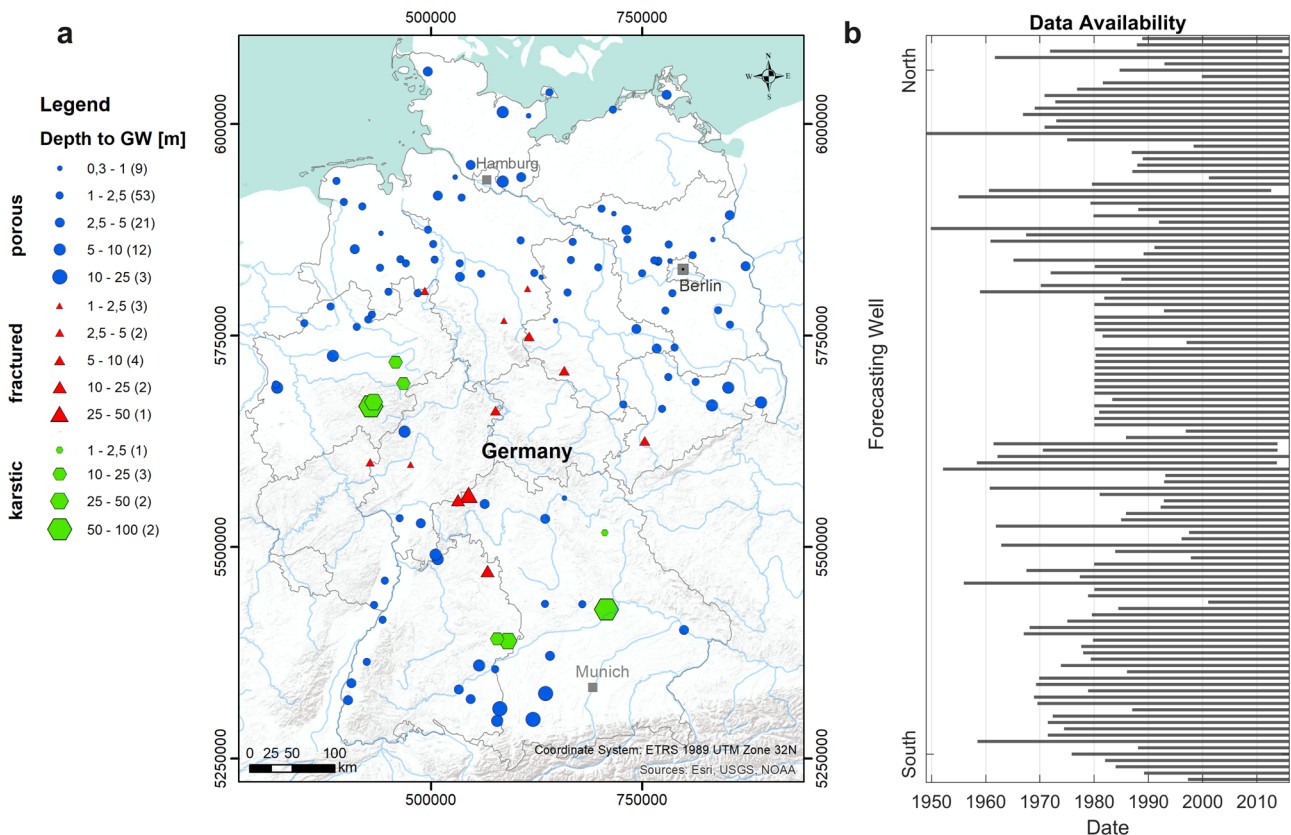

**Fig. 6 Overview map and data availability. a** Position, type of aquifer, and depth to groundwater for each study site. **b** Time series length of all study sites ordered in North–South direction.

recent developments illustrate, the succession of several dry years is much more critical than the overall trend. In such periods, the projected effects accumulate over consecutive years to extremely low groundwater levels, and thus more severe consequences are to be expected. Such longer dry periods are most likely to be averaged out in linear trend analysis, as performed in this study. Nevertheless, we see an increasing frequency of them in all RCP scenarios[40], especially in RCP8.5 and less pronounced in RCP4.5 (Fig. 3). Future research should pay attention to this aspect more intensively. It is also important to highlight that we only model direct climate effects on groundwater levels, and we assume that the basic input–output relationship or system behavior does not change. However, it can most certainly be expected that the system behavior will be influenced by future changes in groundwater extractions, changes in vegetation and land use, as well as surface sealing and other related factors. Groundwater withdrawals, in particular, are expected to increase due to (i) the regionally growing population especially in metropolitan areas (drinking water demand) and (ii) the increasing demand for the industry, energy, and especially irrigated agriculture. As a result, the groundwater level will inevitably drop further, if no active measures such as limitation of withdrawals, avoidance of irrigated agriculture by changing crop types, or even artificial recharge by infiltration, are applied. Despite all these limitations, the results give a good impression of the magnitude of changes to be expected purely due to direct climatic influences.

## Methods

**Data**. We used weekly groundwater level data from 118 different sites, well distributed all over Germany (Fig. 6a). All wells are located in the unconfined, uppermost (thus mostly shallow) aquifers, which are most likely to be subject to direct climate change effects[12]. Greater depths to groundwater are predominantly found in fractured and karstic aquifers. For additional details on the sites please

refer to the supplementary material (Supplementary Table S1). Groundwater level records of all sites show very different lengths (Fig. 6b), from 15 to 67 years, with a median length of 36 years. Data gaps were closed using the information of several related groundwater level time series with highly correlated dynamics derived from an earlier comprehensive cluster analysis based on hydrograph dynamics[41,42]. Alternatively, PCHIP (Piecewise Cubic Hermite Interpolating Polynomial) was used to close smaller data gaps, where no correlated hydrograph information was available. In our dataset, 48 time series had no missing values; another 44 had less than 2% interpolated values. Only very few time series show a higher proportion of interpolated values (11 time series > 4%). More information on interpolated values can be found online in the released dataset.

Input variables for our models are precipitation (P) and temperature (T), thus purely climatic. These variables are widely available and easy to measure both in the past and present, and are also well evaluated in terms of climate projection output. Precipitation serves as a proxy for groundwater recharge, temperature for evapotranspiration. Additionally, the temperature usually shows a distinct annual cycle, which also provides the models with valuable information on seasonality. Since we specifically selected wells with high forecast accuracy in the past (see "Model calibration and evaluation"), we can assume that the groundwater dynamic at these wells is mainly dominated by climate forcings. As long as no fundamental change of the system relations occurs (e.g., newly installed groundwater pumping or severe changes in land use nearby), we can expect reasonable results for our simulations, as we explore only the influence of changing climate and assume other boundary conditions fixed.

Besides the groundwater level data itself, we based our analysis on several datasets. The models were trained using temperature and precipitation data from the HYRAS dataset[43,44], which is a gridded ($5 \times 5$ km²) meteorological dataset based on observed data from meteorological stations ranging from 1951 to 2015. To evaluate the influence of climate change we used RCP scenario data from several selected climate projections that form the so-called core ensemble defined by DWD[36] (Table 1). Depending on the scenario and the considered variable, this ensemble represents 80–90% of the ensemble spread of the possible climate signal within the larger 'reference-ensemble'[36]. The latter, in turn, constitutes all available and quality-assessed projections for Germany. Further, we received the projection data bias-adjusted onto the HYRAS dataset and regionalized it on a $5 \times 5$ km² grid by ref. [35]. For each site, the mean of $3 \times 3$ cells around the cell with the respective well was chosen as input for the simulations, following the best practices by DWD to reduce uncertainty resulting from the grid cell size.

Generally, the used climate projections show a slight increase in precipitation sums and a significant temperature increase of several degrees Celsius for Germany by

**Table 1 Climate projections overview.**

| Scenario | Projections | Abbrev. |
|---|---|---|
| **RCP8.5** | CCCma-CanESM2_rcp85_r1i1p1_CLMcom-CCLM4-8-17 | p1 |
| | ICHEC-EC-EARTH_rcp85_r1i1p1_KNMI-RACMO22E | p2 |
| | MIROC-MIROC5_rcp85_r1i1p1_GERICS-REMO2015 | p3 |
| | MOHC-HadGEM2-ES_rcp85_r1i1p1_CLMcom-CCLM4-8-17 | p4 |
| | MPI-M-MPI-ESM-LR_rcp85_r1i1p1_UHOH-WRF361H | p5 |
| | MPI-M-MPI-ESM-LR_rcp85_r2i1p1_MPI-CSC-REMO2009_v1 | p6 |
| **RCP4.5** | ICHEC-EC-EARTH_rcp45_r1i1p1_KNMI-RACMO22E_v1 | p1 |
| | ICHEC-EC-EARTH_rcp45_r12i1p1_KNMI-RACMO22E_v1 | p2 |
| | ICHEC-EC-EARTH_rcp45_r12i1p1_SMHI-RCA4_v1 | p3 |
| | MOHC-HadGEM2-ES_rcp45_r1i1p1_CLMcom-CCLM4-8-17_v1 | p4 |
| | MPI-M-MPI-ESM-LR_rcp45_r1i1p1_MPI-CSC-REMO2009_v1 | p5 |
| | MPI-M-MPI-ESM-LR_rcp45_r2i1p1_MPI-CSC-REMO2009_v1 | p6 |
| **RCP2.6** | ICHEC-EC-EARTH_rcp26_r12i1p1_CLMcom-CCLM4-8-17_v1 | p1 |
| | ICHEC-EC-EARTH_rcp26_r12i1p1_KNMI-RACMO22E_v1 | p2 |
| | MIROC-MIROC5_rcp26_r1i1p1_CLMcom-CCLM4-8-17_v1 | p3 |
| | MOHC-HadGEM2-ES_rcp26_r1i1p1_KNMI-RACMO22E_v2 | p4 |
| | MPI-M-MPI-ESM-LR_rcp26_r2i1p1_MPI-CSC-REMO2009_v1 | p5 |

For more information on the models please visit https://www.euro-cordex.net/.
Climate projections used in this study and according to abbreviations used throughout the text.

2100[11,37,38], more precise values depending strongly on the considered scenario. For RCP8.5, an input data analysis at the relevant 118 sites of this study showed a consistent annual average temperature increase in all regions of Germany of several degrees Celsius (mostly between 3 and 4 °C). Only very slight spatial patterns emerge, with strongest increases in the South (up to 4.7 °C) and generally slighter increases in the Northwest, probably due to a buffer effect near the coast. For the total annual precipitation, non-significant changes ($p > 0.05$) dominate. The fewer significant changes partly show opposing trends, depending on the projection. One projection shows consistent decreases of mostly −150 mm (max: −367 mm in the far South). Some other projections show increasing precipitation instead (mostly around 100 mm) except for the Northwest, where almost no increases are visible. The southern part shows the strongest possible increases in precipitation, up to 300 mm. Under RCP4.5 the respective input data reveals no spatial pattern in the case of the temperature. Input data shows spatially consistent increases mostly between 1 °C and 2 °C. For the precipitation data, non-significant results dominate. However, the few significant increases show a clear spatial pattern and occur mostly in the South and Northwest, ranging mostly around 100 mm; in the eastern part, we see basically no increasing precipitation. Under RCP2.6 non-significant results are dominating. In terms of the temperature data, however, we find a spatial pattern of slight, yet significant increases (0.5–0.8 °C) in the North and Northeast, as well as for the upper Rhine graben area in the Southwest. For the precipitation only a few significant results occur, showing decreases of about −100 mm, mostly in the Northwest. For map and boxplot representations of these analyses, please refer to the Supplementary Figs. S3–S8.

**Convolutional neural networks (CNNs).** CNNs[45] are commonly used for image recognition and classification tasks but also work well on sequential data, such as groundwater level time series[14]. The CNNs used in this study comprise a 1D-convolutional layer with fixed kernel size (three) and optimized number of filters, followed by a Max-Pooling layer and a Monte-Carlo dropout layer, applying a fixed dropout of 50% to prevent the model from overfitting. This dropout rate is quite high and forces the model to perform very robust training. A dense layer with an optimized number of neurons follows, succeeded by a single output neuron. We used the Adam optimizer for a maximum of 100 training epochs with an initial learning rate of 0.001 and applied gradient clipping to prevent exploding gradients. Early stopping with patience of 15 epochs was applied as another regularization technique to prevent the model from overfitting the training data. Several model hyperparameters (HP) were optimized using Bayesian optimization[46]: training batch-size (16–256); input sequence length (1–52 weeks); the number of filters in the 1D-Conv layer (1–256); the size of the first dense layer (1–256). All models were implemented using Python 3.8[47], the deep-learning framework TensorFlow[48], and its Keras[49] API. Further, the following libraries were used: Numpy[50], Pandas[51,52], Scikit-Learn[53], BayesOpt[46], Matplotlib[54], Unumpy[55], and SHAP[33].

**Model calibration and evaluation.** We used weekly groundwater level time series data of varying lengths (Fig. 6b). To find the best model configuration, we split every time series into four parts: training set, validation set, optimization set, and test set. The test set uses always the 4-year period from 2012 to 2016 (Fig. 7b, s.a. Figure 8a for an example, for a few sites where the time series ended slightly earlier, we shifted the 4-year test set period accordingly). The first 80% of the remaining time series before 2012 were used for training, the following 20% for early stopping (validation set) and

for testing during HP optimization (optimization set), using 10% of the remaining time series each (Fig. 7b). As target function during HP optimization, we chose the sum of Nash–Sutcliffe efficiency (NSE) and squared Pearson r ($R^2$) (compare ref. [14]), the acquisition function is expected improvement. For each model, we used a maximum optimization step number of 150 or stopped after 15 steps without improvement once a minimum of 60 steps was reached. Generally, we scaled the data to [−1,1] and used an ensemble of ten pseudo-randomly initialized models to reduce the dependency towards the random number generator seed. For each of the ten ensemble members, we applied Monte-Carlo dropout during simulation to estimate the model uncertainty from 100 realizations each. We derived the 95% confidence interval from these 100 realizations by using 1.96 times the standard deviation of the resulting distribution for each time step. Each uncertainty was propagated while calculating the overall ensemble median value for final evaluation in the test set (2012–2016). We calculated several metrics to judge the simulation accuracy: NSE, squared Pearson r ($R^2$), absolute and relative root mean squared error (RMSE/rRMSE), as well as absolute and relative Bias (Bias/rBias). Note that we calculate NSE with a long-term mean GWL before the test set instead of the test set mean value. Please see ref. [14] for more details on calculation as the same approach was used. We use almost exclusively wells, at which the models showed a very high forecast accuracy in the test-set (mostly NSE and $R^2$ larger than 0.8, compare to Fig. 7a). Some models were included with slightly lower accuracy (at least NSE and $R^2$ larger than 0.7) to improve the spatial coverage resulting in a set of 118-wells from all over Germany. For additional details on the error measures and HP for all sites please refer to our Supplementary material. Figure 8a shows the model evaluation on the test set exemplarily for one well.

**Model plausibility and interpretability.** To perform groundwater level simulations until 2100 we retrained all models using the defined HP and all data until 2014. Hence, we split the time series only into two parts: 80% for training and 20% for early stopping (Fig. 7b). Afterward, we assessed the model stability and the plausibility of the output values in the extrapolated regime accordingly to ref. [30] by evaluating the model output using artificially altered input data based on historical observed climatology with quadruple precipitation and systematically 5 °C higher temperature (Fig. 8b). As long as the model output does not "blow up" or produce meaningless outputs[30], we can hereby improve confidence in the simulation results when investigating the different RCP scenarios. Models showing implausible behavior in preliminary analyses were not considered for this study. We additionally applied an explainable AI approach to check whether the models have learned correctly in terms of our conceptual understanding of hydrogeological processes. We calculated SHAP[33] values that explain the influence (sign and strength) of every input feature value on the model output (Fig. 8c). Generally, our models showed that the relationship between input and output was captured plausibly. For example, high precipitation inputs (P, red) produce high SHAP values and therefore have a strong positive influence on the model output, which corresponds to our basic understanding of the influence of recharge, leading to increasing groundwater levels. Low or no precipitation (P, blue) has a comparably slight negative influence on GWL, whereas high-temperature inputs (T, red) have a strong negative influence on the model output. Again, this corresponds with our basic understanding of the governing processes, where the high temperature usually means high evapotranspiration, which causes less recharge or even direct groundwater evaporation in some cases. This sounds trivial, however, during

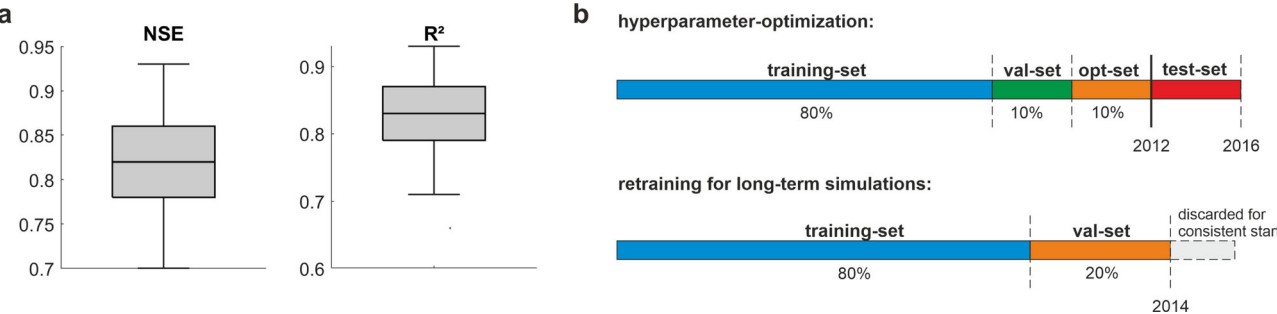

**Fig. 7 Overall model performance and data splitting scheme. a** Model performance of all models for the test-set (2012–2016). **b** Time series splitting scheme for optimization (upper) and retraining (lower).

**Fig. 8 Model performance, plausibility, and interpretability. a** Optimized model evaluation in the past for the test set (2012–2016). **b** Model output under an artificial extreme climate scenario in the past. **c** SHAP Summary plot.

preliminary work for this study, we found that not all models captured these relations correctly, which also partly caused erroneous values in the extrapolated regime (see above). Such models were excluded from the final study. Figure 8 exemplarily summarizes the model evaluation (a) and plausibility checks (b, c) for one well. Respective figures of all other sites are provided in the supplement (Supplementary Figs. S9–S126).

**Evaluation of the projected groundwater levels**. For our simulation results until 2100, we examined the relative development of the mean as well as the 2.5% (lower extreme), and 97.5% (upper extreme) quantile. All were site-specifically calculated

on a yearly basis for each individual projection followed by a linear trend analysis based on Mann–Kendall, and Theil–Sen slope. In doing so, we are able to capture both the range and the individual development of all considered future climate projections. Even though considering yearly values, we applied the 3PW pre-whitening method[56] (implemented in the Mann–Kendall/Python[57] package) to eliminate the remaining first-order autocorrelation before applying Mann–Kendall test and calculating corresponding Theil–Sen slopes. To make comparisons between different sites possible, results are normalized on the individual range of each historic groundwater level time series between the years 2000 and 2014 (start of simulation due to data availability). Even though all climate projections are bias-adjusted on the HYRAS training dataset, they still do not depict the real climate

development for individual years (also historically), which can cause a bias between the end of historical data records and the start of our simulations. We, therefore, investigated the trend of the aforementioned quantities between the start of the simulation and the end in 2100 and did not directly consider the end of the historical records. We examined each groundwater level development using Mann–Kendall linear trend test[58] and derived the relative development in percent from a linear fit using Theil–Sen slope. For Mann–Kendall test we considered a trend significant for $p < 0.05$, and we further provide upper and lower 95% confidence bounds of the Theil–Sen slopes[59] for all significant trends.

## Data availability

The original groundwater level data are available free of charge from the respective local authorities: LUBW, LfU Bavaria, LfU Brandenburg, HLNUG, LUNG Mecklenburg-Western Pomerania, NLWKN, LANUV North Rhine-Westphalia, LfU Rhineland-Palatinate, SMUL, LHW Saxony-Anhalt, and LLUR Schleswig-Holstein. We published the processed groundwater level data including interpolated values based on the previous knowledge[41] with the kind permission of these local authorities: https://doi.org/10.5281/zenodo.4683879. Climate projection data are available on request and free of charge for non-commercial and academic purposes from the German Meteorological Service (DWD).

## Code availability

The code necessary to reproduce our results is available on GitHub.[60]

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

## Acknowledgements

We acknowledge the support and advice by the German Meteorological Service in providing and handling the climate data.

## Author contributions

All authors contributed to the conceptualization of this study. A.W. and T.L. contributed to the methodology, A.W. wrote the software code, performed validation, formal analysis, investigation, visualization, and wrote the original draft. S. B. performed data curation activities. All authors contributed to reviewing and editing the draft. T.L. and S.B. both supervised the work and were involved in project administration.

## Funding

## Competing interests

The authors declare no competing interests.
