## [Peer Review File · Nature Communications]

Deep learning shows declining groundwater levels in Germany until 2100 due to climate changeReviewers' Comments:

Reviewer #1:

Remarks to the Author:

Review of Deep learning shows declining groundwater levels in Germany until 2100 due to climate change by A. Wunsch, T. Liesch, and S. Broda

In this work, the authors attempted to assess the potential impact of future climate on groundwater (GW) levels in 118 wells in Germany. To do this, they trained 1D CNN models using historic weekly GW observations from 1950 to 2015 (after gap filling), and then applied the trained CNN models to predict future GW levels by using projected precipitation and temperature as forcing. Although the motivation is good, I'm concerned with the validity of applying this data-driven approach to future climate scenarios.

1. The known "unknowns" in this case, as the authors mentioned, are "anthropogenic groundwater withdrawals" and "associated with land-use changes". There's plenty of evidence suggesting many contemporary global hydrological models couldn't simulate current-day human intervention very well, not to mention future scenarios. For example, a study by Scanlon et al. (2018) compared the simulated groundwater storage trends to that observed by GRACE satellites for a large number of global river basins and noticed large discrepancies between simulated and observed trends. In the future, as the authors mentioned "the impact of these factors will be exacerbated as water demand increases..." (L54). So the compounding effect caused by climate change and human intervention on GW may not be a linear one. Thus, using P and T to project future GW trend using data driven method is generally not reliable. For the same reason, I also question the main premise on L95-96, "...due to high prediction accuracy in the past, the selected sites are unlikely to be under the influence of strong groundwater withdrawals or comparable effects..." As I elaborate in the next bullet, a good performance on historical data is not a guaranty for future performance.

2. It is well known that data driven methods are not good at extrapolation. In other words, these methods aren't good at predicting instances that are not seen during training and they are not good at predicting nonstationary time series. If for some reason, there's a change in trend or there are huge spikes that are out of the training data range, the data-driven methods will usually fail. As a case in mind, Sun et al. (2020) trained numerous machine learning models to predict total water storage in the U.S. However, significant wetting trends occurred in several basins during the "future" phase. The authors showed that the data-driven methods couldn't capture the trend change well.

3. Methodology wise, I'm concerned with using 1D CNN for time series forecasting, especially when dealing with long sequences (52 weeks). This is because CNN has a fixed reception field (in their work, the authors used a fixed kernel size 3), which cannot capture multiscale temporal correlations very well. Based on my own experience, LSTM would be a much better choice in terms of forecasting accuracy on time series with long memory.

4. On data analysis part, data gap filling is a huge issue and almost deserves a separate analysis on its own. Here Figure 6 shows data availability is pretty limited pre-1980. Any interpolation will add artifacts to the time series. The authors treatment of this issue was surprisingly cursory. It's not clear how the authors assessed the quality of gap filling.

5. The authors showed temperature is a dominant predictor, which is not new as GW level in humid regions is generally dominated by seasonality. However, this is probably only valid in Germany, not valid in many other arid and semiarid regions that depend more on GW as a critical water supply. Thus, a more meaningful task would be to predict inter-annual GW change instead of full signal that's dominated by seasonal variations and uncertain trend.

References:

Scanlon, B. R., Zhang, Z., Save, H., Sun, A. Y., Schmied, H. M., Van Beek, L. P., ... & Bierkens, M. F. (2018). Global models underestimate large decadal declining and rising water storage trends relative to GRACE satellite data. *Proceedings of the National Academy of Sciences*, 115(6), E1080-E1089.

Sun, A. Y., Scanlon, B. R., Save, H., & Rateb, A. (2020). Reconstruction of GRACE Total Water Storage Through Automated Machine Learning. *Water Resources Research*, e2020WR028666.

Rodell, M., Famiglietti, J. S., Wiese, D. N., Reager, J. T., Beaudoin, H. K., Landerer, F. W., & Lo, M.

H. (2018). Emerging trends in global freshwater availability. *Nature*, 557(7707), 651-659.

Reviewer #2:

Remarks to the Author:

Wunsch et al. present in their manuscript "Deep learning shows declining groundwater levels in Germany until 2100 due to climate change" some interesting results on the potential groundwater response on climatic changes. However, there are strong limitations which are currently not addressed and prevent results to be useful. To make valid statements on future groundwater levels it will be necessary to analyse much more the different RCPs and the different sources of uncertainty.

1. The authors state they use only projections based on RCP 8.5 (l. 87). This is a major constraint and prevents to derive any general future predictions. Usually, the different RCP are used to analyse the bandwidth of potential future changes, therefore, using only the most extreme one (leading to the strongest changes and trends) requires some strong reasoning. Unfortunately, any reasoning or discussion of this point is completely missing in the manuscript. Without considering different RCPs the authors cannot claim to present valid predictions for 2100. However, for most parts of the manuscript this important limitation does not become clear. E.g. many of the results are written like forecasts ("heads will probably decrease", "expected to show increased values", etc.). Also, the title exaggerates the findings without mentioning the constraints.

2. The authors present many results for the selected set of 6 different climate models. However, the climate model is only one source of uncertainty and not necessarily the most important one. Other sources of uncertainty which are probably relevant in this context include: groundwater model uncertainty (from the supplements it is evident that model performance differs between sites and that there are larger uncertainties for the simulation of extremes); scaling uncertainty (grid of 5x5km vs. borehole); statistical analysis uncertainty (limitations of MK-Test and trend analysis), emission scenario uncertainty (see above); etc. While the authors quantify and discuss climate model uncertainty, all the other uncertainties are neglected. However, without a reliable uncertainty analysis results are not useful and cannot depict the expectable changes of groundwater levels in Germany until 2100.

Minor points:

- l. 89 "represent 80% of the possible future climate signal" -> this high percentage is puzzling given that only one RCP is used in this study and hence a very small proportion of possible future climate signals is covered by the runs.
- l. 106: Is a linear trend an appropriate functional form to describe the change? For example, in case the real trend at a station is rather exponential, the linear trend could give values that deviate for 2100 quite a bit. In general, the fitted values at the end of the timeseries quite often deviate from actual values.
- ll. 113 f.: unit of mm/y not clear. Seems like a rate of change (i.e. the slope of the trend line), but I guess that is not meant here.
- ll. 140 ff.: All these results are focussed on annual percentiles, correct? Above you mention the different water users and potential water conflicts, also different climatic changes within the year are mentioned -> did you also look on groundwater trends for the different seasons? Based on Figure 3 I can already guess that there are some relevant seasonal differences. These can be also very relevant for water management.
- Figure 3: From my perspective this figure contains way too many plots which are too small to be readable.
- l. 454: Probability values of Mann-Kendall are only valid in case of no autocorrelation which is usually not the case for groundwater records. Were autocorrelations calculated and timeseries pre-whitened?

Reviewer #3:
Remarks to the Author:
Comments:

This paper is of great interest not only from a scientific point of view but also for practitioners, as questions about our future water resources are piling up. It is an exciting contribution to study the future climatic impacts on groundwater quantity in the future. Referring to the text I have the following comments and questions:

In my opinion, the title is somewhat misleading, as the paper only focuses on the worst-case scenario (RCP8.5) and ignores all other future projections. Current studies show that even with a 'business as usual' development - regarding CO₂-emissions - the bandwidth of projected results will be partly below the range of the RCP8.5 projections, which means the effects for groundwater fluctuations is quite smaller. Therefore, it makes sense to mention the used RCP scenario in the paper title. This points out to the reader right from the start that only parts of the available climate projections was used.

Line 12ff.: ...RCP8.5 scenario ... represent 80% of the bandwidth...
From my point of view the following details are missing in the paper: Why only RCP8.5 projections are chosen? What does it mean, when 80% of the bandwidth is used? (Here, for example, the authors could refer to the IPCC classification of likelihoods).

Line 28ff.: ...on groundwater and springs...
It is true that groundwater plays a crucial role in some parts of Germany (and also on the national level in the whole). However, there are also federal states that increasingly use surface water. Perhaps this circumstance should therefore also be mentioned in order to differentiate the significance of the result on a regional level.

Line 34ff.: ...less than 2% of the total withdrawal volume...
Does this value apply to an average in Germany, or is it a regional figure that applies to all federal states?

Line 41ff.: ...of several degrees...by 2100.
Here it would be better to use the original literature, where the data were first described, such as by EURO-CORDEX or the Reklies-De project.

Line 43ff.: For Europe...
Why do the authors go from Germany to Europe, only to return to Germany later?

Line 43ff.: snow dominated regions...
What role does that play for Germany as a whole. I think that this is only relevant for the South.

Line 43ff.: ...unconfined shallow aquifers...
What about regions characterised by fractured aquifers or karstic aquifers? You cannot simply ignore the different aquifers with their different characteristics, which are totally different to shallow porous aquifers.

Line 69ff.: ...declines up to 10 m close to the Alps...
How big was the model error in this study? How good were the statements in relation to the prevailing groundwater thickness? What about areas with aquifers less than 10 m thickness?

Line 81ff.: ...respective uppermost unconfined aquifer...
How representative are the selected wells and springs for the whole of Germany or selected groundwater landscapes?

Line 87ff.: ...downscaled 5 x 5 km²...

How do this resolution and the size of the catchment of selected wells/springs fit together? Was a weighted allocation carried out?

Line 103ff.: ...Germany by 2100...

The references for the climatic information used are not primary references.

Line 103ff.:exact values....

If results from climate projections are used, there are no exact values but only bandwidths of the entire ensemble.

Line 118ff.:...under the RCP8.5 scenario...

From my point of view, it would be good to briefly draw a reference to the other scenarios in order to be able to better classify the results. For example, by pointing out that the approach used shows the greatest possible impact, whereas small effects are to be expected when other RCPs are used.

Line 141ff.:...in 2100...

What does this time indication mean? Since it is a 30-year average, different time periods are possible, such as 2071-2100 or similar. Please specify exactly.

Line 142ff.:...the simulation (2014)....

Why was 2014 chosen as the start of the simulation? Is this for technical or other practical reasons?

Line 151 and others:.....significant trend...

How was significant defined? When is a trend called significant? Since there are different approaches for testing the significance of data, further information would be useful.

Line 216:...(2070-2100)...

I think it should be 2071-2100.

Line 243:... We do not findincreasing mean trends...

How does this fit with the statement that the amount of precipitation increases in the year?

Line 272:... Even fewer significant shift...

Are there any classification steps for significance?

Line 284 and ff:... that temperature is mainly the driving factor for declining groundwater levels...

It should be explicitly mentioned here that the results only apply to shallow aquifers. It would also make sense to define what is meant by "shallow aquifer". Finally, it could also be helpful to address the issue of the behavior of different aquifer types in the discussion.

Response to Reviewers

We thank the reviewers for their comprehensive reviews, and their appreciative and constructive comments. We are happy to read that our paper is described as “of great interest” and used the constructive criticism to substantially improve the manuscript. In the following, please find our answers (red) on the review comments (black). The line numbers in the review comments refer to the originally submitted manuscript, the line numbers in our answers refer to the revised version.

Decision on Nature Communications manuscript NCOMMS-21-14445

Dear Mr Wunsch,

Thank you again for submitting your manuscript "Deep learning shows declining groundwater levels in Germany until 2100 due to climate change" to Nature Communications. We have now received reports from 3 reviewers and, after careful consideration, we have decided to invite a major revision of the manuscript.

As you will see from the reports copied below, the reviewers raise important concerns. We find that these concerns limit the strength of the study, and therefore we ask you to address them with additional work. Without substantial revisions, we will be unlikely to send the paper back to review. In particular, reviewers agree the current approach limits the robustness of the conclusions. To move forward with a revised manuscript, additional analyses using other RCP scenarios is needed. We also agree with Reviewer #2 that a full accounting of sources of uncertainty would strengthen the utility of the results. While we do not require a change in methods, per Reviewer #1's suggestion for the use of a long short-term memory network, we urge you to provide an expanded justification of the choices made in this analysis and representativeness of the selected wells (Reviewer #3).

If you feel that you are able to comprehensively address the reviewers' concerns, please provide a point-by-point response to these comments along with your revision. Please show all changes in the manuscript text file with track changes or colour highlighting. If you are unable to address specific reviewer requests or find any points invalid, please explain why in the point-by-point response.

REVIEWER COMMENTS

Reviewer #1 (Remarks to the Author):

Review of Deep learning shows declining groundwater levels in Germany until 2100 due to climate change by A. Wunsch, T. Liesch, and S. Broda

In this work, the authors attempted to assess the potential impact of future climate on groundwater (GW) levels in 118 wells in Germany. To do this, they trained 1D CNN models using historic weekly GW observations from 1950 to 2015 (after gap filling), and then applied the trained CNN models to predict future GW levels by using projected precipitation and temperature as forcing. Although the motivation is good, I'm concerned with the validity of applying this data-driven approach to future climate scenarios.

Thank you very much for your assessment of the manuscript. We understand your concerns and try to answer in detail to the following statements.

1. The known “unknowns” in this case, as the authors mentioned, are “anthropogenic groundwater withdrawals” and “associated with land-use changes”. There’s plenty of evidence suggesting many contemporary global hydrological models couldn’t simulate current-day human intervention very well, not to mention future scenarios. For example, a study by Scanlon et al. (2018) compared the simulated groundwater storage trends to that observed by GRACE satellites for a large number of global river basins and noticed large discrepancies between simulated and observed trends. In the future, as the authors mentioned “the impact of these factors will be exacerbated as water demand increases...” (L54). So the compounding effect caused by climate change and human intervention on GW may not be a linear one.

Thank you for this comment. We completely agree with your assessment of the future development. We cannot account for land use changes, increased anthropogenic pumping and other such factors. Because of these reasons we do not state to project the real groundwater level development, but only the direct climatic influence under current boundary conditions. We have now better highlighted this aspect (L. 108-115, 401ff). Until now it remained unclear, what the pure climatically driven development of groundwater for Germany might be, because we do not have a very intuitive development of the climatic key forcings such as precipitation and temperature. T increases clearly but also does P, depending on the region. We try to answer which influence dominates the development (L42f).

Thus, using P and T to project future GW trend using data driven method is generally not reliable. For the same reason, I also question the main premise on L95-96, “...due to high prediction accuracy in the past, the selected sites are unlikely to be under the influence of strong groundwater withdrawals or comparable effects...” As I elaborate in the next bullet, a good performance on historical data is not a guaranty for future performance.

In our opinion, using P and T as inputs is reliable because – as elaborated above – we calculate only the climatic influence under existing boundary conditions. We therefore respectfully disagree to the fact that “using P and T to project future GW trend using data driven method is generally not reliable”. For high prediction accuracy in the past it is necessary that a very strong relationship between climate variables and groundwater level exists for a specific site. If other factors were dominant, the model would produce less accurate results in the past. Concerning the performance in the future, we agree that there is never a guarantee of good performance, not for these models nor any other (e.g. physically-based) models. To account for this, we took several measures to increase the confidence in our models and the produced results (high performance in the past, high dropout rate, SHAP values (e.g. L 523ff.), extrapolation behavior (e.g. 514ff.) etc.). Please see also our elaborations for the next bullet point.

2. It is well known that data driven methods are not good at extrapolation. In other words, these methods aren’t good at predicting instances that are not seen during training and they are not good at predicting nonstationary time series. If for some reason, there’s a change in trend or there are huge spikes that are out of the training data range, the data-driven methods will usually fail. As a case in mind, Sun et al. (2020) trained numerous machine learning models to predict total water storage in the U.S. However, significant wetting trends occurred in several basins during the “future” phase. The authors showed that the data-driven methods couldn’t capture the trend change well.

Thank you for pointing out this important aspect. We agree, the data driven models start to fail at some point of extrapolation. However, we see several reasons that this is not the case for our models using future climate scenario data.

First, we have carefully evaluated our models and performed the mentioned plausibility checks which already used unrealistically high values for P (x4) and T (+5°C) in the past (L. 514ff.). Even for those, the models did not completely fail. Of course, absolute values were not realistic,

but the models still produced – at least visually – plausible output patterns in the past that correspond to our conceptual understanding. (compare Fig. 9b).

Second, we do not perform an extrapolation in a classical sense with data out of the (absolute) training range. The changes in future climate patterns we see (e.g. increasing temperatures) are changes in mean values and absolute future values usually still are in the range of the training data (e.g. in the future we might see more regular temperatures above 30°C for a certain location, but we have seen these temperatures also in the past, just less often). We therefore hold the opinion that we do not leave the data manifold in the future. As long as we are confident that our models learn the input output relation in a correct manner (conceptually checked by our explainable AI - SHAP value approach), we argue that we can assume that there is meaning in the forecasted values. We have added and discussed this aspect to the newly added uncertainty section and hope that it becomes clearer now (L 535-358)

3. Methodology wise, I'm concerned with using 1D CNN for time series forecasting, especially when dealing with long sequences (52 weeks). This is because CNN has a fixed reception field (in their work, the authors used a fixed kernel size 3), which cannot capture multiscale temporal correlations very well. Based on my own experience, LSTM would be a much better choice in terms of forecasting accuracy on time series with long memory.

Thank you for this comment, and we completely understand your concerns. For the same reason we have conducted a study (Wunsch et al. 2021, see below), where we explore the suitability of different model types on groundwater level prediction. We have shown, that 1D-CNNs mostly outperform LSTMs in case of groundwater level forecasting, which is the reason we used a similar approach here. We agree, that in theory LSTMs are probably more suited, but besides the mentioned performance differences, in our experience, LSTMs are less stable. Moreover, the receptive field of each individual kernel is indeed three, but we use a large number of kernels, where each of them can detect other features in the complete input sequence of up to one year (length is optimized for each site). We have extended the justification of the choice of methods in Lines 59-67.

4. On data analysis part, data gap filling is a huge issue and almost deserves a separate analysis on its own. Here Figure 6 shows data availability is pretty limited pre-1980. Any interpolation will add artifacts to the time series. The authors treatment of this issue was surprisingly cursory. It's not clear how the authors assessed the quality of gap filling.

Thank you for pointing out this important aspect. We want to point out, that the data availability is indeed limited before 1980 and we think there is a misunderstanding concerning this figure. To clarify, we did not extrapolate the initial length of the time series. The time series length is as shown in the manuscript.

Concerning interpolation of data gaps itself, we used mostly prior knowledge of hydrographs with similar dynamics. This way, we were able to close gaps using the course of a similar hydrographs, not showing a gap in the respective period. Where this was not possible or did not yield plausible results, we used PCHIP interpolation. As part of a previous project, the similarity of the dynamics of several thousand hydrographs all over Germany were analyzed (Wunsch & Liesch 2020), unfortunately the report is only available in German. The results from this report form the basis of our preprocessing strategy. We have added a clarifying statement to the text (Lines 421-428). A paper, which demonstrates the methodology on a subregion of Germany was recently published in Water Resources Management (Wunsch et al. 2021)

Concerning your statement of added artifacts to the time series, we agree, but draw a different conclusion. Based on our knowledge (examples follow) of length and proportion of interpolated data gaps, we think that the added artifacts are neglectable.

Of all 118 time series, 48 had no missing values, other 44 had less than 2% interpolated values (about 20 values for a hypothetical time series of 20 years of data). Only very few time series show a higher proportion of interpolated values (11 time series > 4%). We have published the complete groundwater dataset (see below) and please feel free to check for each site individually the amount of interpolated data and which method was used. To illustrate, in the following, a figure from the published data set is shown for the time series with by far the largest proportion of interpolated values (14%). As you can see, mostly shorter sections had to be interpolated, which do not strongly influence the overall dynamics, because no high frequency changes can be observed overall. Around 2005 (for example) a larger section has been interpolated based on information of highly correlated neighboring time series and we also yield a very plausible pattern here.

The following figure is from the electronic appendix of Wunsch&Liesch (2020) and shows some of the correlated time series which the interpolation shown above was based on. As you can see, we find very similar dynamic patterns and we can use this information to close data gaps with comparably high reliability.

Overall, we hold believe that the interpolation has no negative effect on the result. We have now extended the section on gap filling in the revised manuscript (Lines 421-4289

References:

- Wunsch, A. & Liesch, T. *Entwicklung und Anwendung von Algorithmen zur Berechnung von Grundwasserständen an Referenzmessstellen auf Basis der Methode Künstlicher Neuronaler Netze*. 191

https://www.bgr.bund.de/DE/Themen/Wasser/Projekte/laufend/F+E/Mentor/mentor-abschlussbericht-I.pdf?__blob=publicationFile&v=2 (2020)

- Wunsch, A., Liesch, T. & Broda, S. Feature-based Groundwater Hydrograph Clustering Using Unsupervised Self-Organizing Map-Ensembles. *Water Resour Manage* (2021). <https://doi.org/10.1007/s11269-021-03006-y>
- Wunsch, A., Liesch, T. & Broda, S. Weekly groundwater level time series dataset for 118 wells in Germany. (2021) doi:[10.5281/ZENODO.4683879](https://doi.org/10.5281/ZENODO.4683879).

5. The authors showed temperature is a dominant predictor, which is not new as GW level in humid regions is generally dominated by seasonality. However, this is probably only valid in Germany, not valid in many other arid and semiarid regions that depend more on GW as a critical water supply. Thus, a more meaningful task would be to predict inter-annual GW change instead of full signal that's dominated by seasonal variations and uncertain trend.

Thank you very much for this interesting idea of predicting change instead of the actual GWL values. We might incorporate this in our future work. For now, we find a basic problem in predicting changes instead of the full signal. When using the full signal, we are able to judge if at least visually our model produces meaningful outputs. However, when simulating inter-annual changes, we get a result, which is harder to judge, because even though each timestep might yield plausible results, when translating back into a time series, we are significantly biased by cumulative errors.

To illustrate this, we show in the following example the translation of predicted changes into a groundwater level time series (not inter-annual changes but on a weekly basis, but overall a similar problem):

This is a time series, for which a highly accurate forecast of the groundwater levels of these two years has been produced by our models (available in the Supplement). In this case, the mere simulation result (not shown), which is changes from timestep to timestep, is not so bad either. However, when translating the changes back into a time series, we are forced to cumulate results from prior steps, which also cumulates prior errors. A correction is only possible if the ground truth (groundwater levels) is known for the test set, which is not until 2100. At the moment, we see no solution to this problem. Therefore, for this specific study, it exceeds the currently possible scope.

Concerning the validity of the results; yes of course, this is only valid for Germany, and even there with all uncertainties and limitations discussed. We do not claim a broader validity or transferability.

Thank you for this compilation of literature. It helped us to illustrate the raised concerns.

References:

Scanlon, B. R., Zhang, Z., Save, H., Sun, A. Y., Schmied, H. M., Van Beek, L. P., ... & Bierkens, M. F. (2018). Global models underestimate large decadal declining and rising water storage trends relative to GRACE satellite data. *Proceedings of the National Academy of Sciences*, 115(6), E1080-E1089.

Sun, A. Y., Scanlon, B. R., Save, H., & Rateb, A. (2020). Reconstruction of GRACE Total Water Storage Through Automated Machine Learning. *Water Resources Research*, e2020WR028666.

Rodell, M., Famiglietti, J. S., Wiese, D. N., Reager, J. T., Beaudoing, H. K., Landerer, F. W., & Lo, M. H. (2018). Emerging trends in global freshwater availability. *Nature*, 557(7707), 651-659.

Reviewer #2 (Remarks to the Author):

Wunsch et al. present in their manuscript “Deep learning shows declining groundwater levels in Germany until 2100 due to climate change” some interesting results on the potential groundwater response on climatic changes. However, there are strong limitations which are currently not addressed and prevent results to be useful. To make valid statements on future groundwater levels it will be necessary to analyse much more the different RCPs and the different sources of uncertainty.

1. The authors state they use only projections based on RCP 8.5 (l. 87). This is a major constraint and prevents to derive any general future predictions. Usually, the different RCP are used to analyse the bandwidth of potential future changes, therefore, using only the most extreme one (leading to the strongest changes and trends) requires some strong reasoning. Unfortunately, any reasoning or discussion of this point is completely missing in the manuscript. Without considering different RCPs the authors cannot claim to present valid predictions for 2100. However, for most parts of the manuscript this important limitation does not become clear. E.g. many of the results are written like forecasts (“heads will probably decrease”, “expected to show increased values”, etc.). Also, the title exaggerates the findings without mentioning the constraints.

Thank you for pointing out these aspects. We now have additionally included RCP4.5 and RCP2.6 in our analyses and have adapted the manuscript accordingly. See especially Lines 226ff and Figures 2 and 3. We admit that our title was a little bit misleading, given that we only investigated RCP8.5. Given our additional analyses for RCPs 2.6 and 4.5 and after careful consideration, we think that the title is adequate now, and does not need to be changed. With the analyses performed, we find declining groundwater levels for all RCP scenarios considered. For most of the wells there are already at least slightly negative trends under RCPs 2.6 and 4.5.

Moreover, we have modified our wording throughout the manuscript and hope that it now better communicates the constraints and limitations, as well as sounds less strongly like as we present precise forecast results.

2. The authors present many results for the selected set of 6 different climate models. However, the climate model is only one source of uncertainty and not necessarily the most important one. Other sources of uncertainty which are probably relevant in this context include: groundwater model uncertainty (from the supplements it is evident that model performance differs between sites and that there are larger uncertainties for the simulation of extremes); scaling uncertainty (grid of 5x5km vs. borehole); statistical analysis uncertainty (limitations of

MK-Test and trend analysis), emission scenario uncertainty (see above); etc. While the authors quantify and discuss climate model uncertainty, all the other uncertainties are neglected. However, without a reliable uncertainty analysis results are not useful and cannot depict the expectable changes of groundwater levels in Germany until 2100.

Thank you for pointing out this weakness of our manuscript. We have strengthened our results, as we now have taken measures to take care of some mentioned sources of uncertainty. Our analysis now considers different RCP scenarios (RCP2.6, RCP4.5 and RCP 8.5), each with an ensemble of different climate models. We also have included additional statements (e.g. Lines 347-367) to further discuss the uncertainty sources in our manuscript. Further, a newly calculated uncertainty (95% confidence interval) derived from Theil-Sen slopes is now included into the presentation of the results to account for the statistical test uncertainty (e.g. Figure 2 and Figure 3). Scaling uncertainty due to the differences between a single location and the grid cell sizes are certainly present. By achieving high performance in the past using training data in the same grid resolution we can assume that this influence is not severe. We further changed the data selection strategy for the climate projections, by now using 3x3 grid cells instead of 1x1 (directly at the location of each site) (L 363 and 453). We hereby follow the recommendations of the German Meteorological Service to account for larger scale atmospheric processes that usually do not scale to a single scale of the used data.

Minor points:

- I. 89 “represent 80% of the possible future climate signal” -> this high percentage is puzzling given that only one RCP is used in this study and hence a very small proportion of possible future climate signals is covered by the runs.

Thank you for pointing out this ambiguous wording. We have modified the respective statement to make the meaning clearer (L 95-99). Further, by analyzing also other RCPs the context should also be better understandable.

- I. 106: Is a linear trend an appropriate functional form to describe the change? For example, in case the real trend at a station is rather exponential, the linear trend could give values that deviate for 2100 quite a bit. In general, the fitted values at the end of the timeseries quite often deviate from actual values.

We think that a linear trend analysis is appropriate, because for our simulation results we do not find exponential trends but rather sections of successive years with stronger/weaker changes. It would be hard to grasp and interpret such periodicities, additionally the linear trend analysis considers the whole simulation period of >80y, which circumvents problems of interpreting shorter than 30y periods (which is not recommended).

- II. 113 f.: unit of mm/y not clear. Seems like a rate of change (i.e. the slope of the trend line), but I guess that is not meant here.

Thank you for pointing out. As we are speaking of the total annual precipitation the unit is indeed mm per year. We rewrote this part of the text, but nevertheless removed “per year” from similar statements (Lines 116ff.).

- II. 140 ff.: All these results are focussed on annual percentiles, correct?

Yes, this is correct. The results are all on annual percentiles.

Above you mention the different water users and potential water conflicts, also different climatic changes within the year are mentioned -> did you also look on groundwater trends for the different seasons? Based on Figure 3 I can already guess that there are some relevant seasonal differences. These can be also very relevant for water management.

Thank you for this suggestion, this would be an interesting extension of our analysis. However, we did not analyze this aspect, because we think that this would exceed the scope of the current study and we already hardly are able to present all results of the current analyses, especially given the additional RCP scenarios that are now included. We will take this suggestion into account for future research.

• Figure 3: From my perspective this figure contains way too many plots which are too small to be readable.

Thank you. We have restructured this figure completely and increased the font size in the new version. We hope this resolves the readability problems.

• l. 454: Probability values of Mann-Kendall are only valid in case of no autocorrelation which is usually not the case for groundwater records. Were autocorrelations calculated and timeseries pre-whitened?

Yes, you are right. MK-Test should not be applied for seasonal data. However, we perform a trend analysis on annual values (e.g. the annual mean), which means that the autocorrelation from typical groundwater seasonality (per year) is not part of the analysis. Therefore, no change in methodology is needed here. To clarify, we have further elaborated this aspect in Line 544ff.

Reviewer #3 (Remarks to the Author):

Comments:

This paper is of great interest not only from a scientific point of view but also for practitioners, as questions about our future water resources are piling up. It is an exciting contribution to study the future climatic impacts on groundwater quantity in the future. Referring to the text I have the following comments and questions:

In my opinion, the title is somewhat misleading, as the paper only focuses on the worst-case scenario (RCP8.5) and ignores all other future projections. Current studies show that even with a 'business as usual' development - regarding CO₂-emissions - the bandwidth of projected results will be partly below the range of the RCP8.5 projections, which means the effects for groundwater fluctuations is quite smaller. Therefore, it makes sense to mention the used RCP scenario in the paper title. This points out to the reader right from the start that only parts of the available climate projections were used.

Thank you for this comment. We agree that our title was a little bit misleading, given that we only investigated RCP8.5. We now additionally show results for RCPs 2.6 and 4.5 and after careful consideration, we think that the title is adequate now, and does not need to be changed. Our analyses show that for all three RCPs considered, declining groundwater level trends can be found. Even under RCPs 2.6 and 4.5 most of the wells show a slight declining trend.

1. Line 12ff.: ...RCP8.5 scenario ... represent 80% of the bandwidth....
From my point of view the following details are missing in the paper: Why only RCP8.5 projections are chosen? What does it mean, when 80% of the bandwidth is used? (Here, for example, the authors could refer to the IPCC classification of likelihoods).

Thank you for pointing out. We applied major changes to our manuscript and included also RCP Scenarios 2.6 and 4.5 into our study (see especially Lines 226ff., Figures 2 and 3). We also clarified what we meant by bandwidth (ensemble spread) (L 96, 449). We still do not use the IPCC classification of likelihoods as this is not what was meant here.

2. Line 28ff.: ...on groundwater and springs...

It is true that groundwater plays a crucial role in some parts of Germany (and also on the national level in the whole). However, there are also federal states that increasingly use surface water. Perhaps this circumstance should therefore also be mentioned in order to differentiate the significance of the result on a regional level.

Yes, on national scale groundwater indeed plays a crucial role (regarding drinking water supply – what is probably meant in the comment). There are of course local and sometime regional differences, but we do not aim to resolve this aspect on such a spatial scale. However, even for surface water, which is strongly interconnected to groundwater via baseflow, the relevance of these results is still high. As shown by de Graaf et al. (2019) (also cited in the manuscript) even a small decrease of 10cm can have severe consequences for baseflow of rivers in northern Germany. Moreover, groundwater is not only relevant regarding drinking water supply, but also for groundwater dependent ecosystems, and therefore plays a crucial role in general. We therefore do not think, that a relativization is necessary at this point.

Reference:

de Graaf, I. E. M., Gleeson, T., (Rens) van Beek, L. P. H., Sutanudjaja, E. H. & Bierkens, M. F. P. Environmental flow limits to global groundwater pumping. *Nature* 574, 90–94 (2019).

3. Line 34ff.: ...less than 2% of the total withdrawal volume...

Does this value apply to an average in Germany, or is it a regional figure that applies to all federal states?

According to the source cited, which is the Federal Environment Ministry (UBA, Umweltbundesamt), this is a number derived from DESTATIS (Federal Statistical Office) data and an overall average for whole Germany.

4. Line 41ff.: ...of several degrees....by 2100.

Here it would be better to use the original literature, where the data were first described, such as by EURO-CORDEX or the Reklies-De project.

Thank you for this hint. In the further course of the text, we exchanged the literature and the original literature is now cited in Lines 98-99. At this specific point in the text we talk about an analysis of water availability. We therefore think that it is adequate to stick with the cited literature.

5. Line 43ff.: For Europe...

Why do the authors go from Germany to Europe, only to return to Germany later?

Thank you. We have adapted the respective sentences and hope it is easier to follow now. (L38ff.).

6. Line 43ff.: snow dominated regions...

What role does that play for Germany as a whole. I think that this is only relevant for the South.

Thank you. We have adapted the respective sentences to better point out that this is only relevant for the South. (L 46f.).

7. Line 43ff.: ...unconfined shallow aquifers...

What about regions characterised by fractured aquifers or karstic aquifers? You cannot simply ignore the different aquifers with their different characteristics, which are totally different to shallow porous aquifers.

Thank you for raising this important concern. We have also included a smaller number of wells in fractured and karstic aquifers. Please check the "Data" section in the "Methods" part of the manuscript for more details on their positioning. Further, we have limited our analysis to the uppermost aquifer at each site, which often happens to be a porous aquifer. In the overall available data that we selected our wells from, the number of wells in fractured and karstic aquifers is both generally, but especially for the uppermost aquifer, by far smaller than the number of wells in porous aquifers. In the end there was only this small number of fractured and karstic aquifer wells that met our rigorous pre-selection criteria. We would have liked to include more of these, but it was just not possible. From a relevance point of view, you are right, we cannot neglect other aquifers, still, shallow porous aquifers are probably the most important ones when it comes to groundwater extractions or water availability, due to the larger volumes that are available there, and also regarding e.g. water availability for vegetation/groundwater dependent ecosystems.

8. Line 69ff.: ...declines up to 10 m close to the Alps...

How big was the model error in this study?

we have chosen models that fit the data with high accuracy for a test period in the past, see e.g. Fig. 9a. However, the individual model error varies from site to site and can be looked up in the Supplementary. The model uncertainty based on different realizations (derived from Monte Carlo dropout, no uncertainty from inputs included) is usually very small. We hope this answers your question.

8.1 How good were the statements in relation to the prevailing groundwater thickness?

We are sorry, but we usually have no information about this locally and we have not investigated this for the same reason.

8.2 What about areas with aquifers less than 10 m thickness?

Good point, thank you. In principle it is possible that the decline is stronger than physically possible. However, such knowledge is not included into the model, nor in the interpretation, since, as mentioned above, we lack this information.

9. Line 81ff.: ...respective uppermost unconfined aquifer...

How representative are the selected wells and springs for the whole of Germany or selected groundwater landscapes?

Representativity is difficult to judge. We have no possibility to check if a single well is representative for a whole groundwater landscape. At least porous aquifers (which are the most) are more important for GW availability in Germany than the other types (with regional differences, of course). Our selection is thus not representative for all areas, but probably for the majority/for the most important ones. Moreover, it is important to emphasize that the results are not suitable for regionalization.

In Wunsch&Liesch (2020) (Report in German), we performed a comprehensive cluster analysis of groundwater dynamics throughout Germany. Based on these cluster results, we already performed interpolation of data gaps (See our answer on question 4 of Reviewer #2). The only thing we can say is that our 118 wells originate from 52 different clusters, which in total comprise time series of more than 2600 wells. However, we cannot directly draw a conclusion on representativeness from this number, because not all clusters are as

homogenous as the example shown above, nor are all of our 118 wells similarly representative for their whole cluster. Nevertheless, this is an indicator that our wells represent the dynamics of a certain number of other wells, too. However, this is generally vague and we therefore refrain from adding this to our manuscript.

10. Line 87ff.: ...downscaled 5 x 5 km²...

How do this resolution and the size of the catchment of selected wells/springs fit together? Was a weighted allocation carried out?

In the original submission directly the grid cell, which the site lies in was selected. We have changed our approach and now follow the best practice recommendations of DWD by taking the mean of 9 (3x3) cells (L 363,453), with the groundwater well grid cell in the middle. We did not include springs in our dataset, groundwater wells mostly do not have a well-defined catchment area. Thus, we stuck with the 9-cell-mean approach.

11. Line 103ff.: ...Germany by 2100...

The references for the climatic information used are not primary references.

Thank you. We have now corrected these references. (L 98,99)

12. Line 103ff.:exact values....

If results from climate projections are used, there are no exact values but only bandwidths of the entire ensemble.

Yes, you are right. What we meant was "more precise" values. We have corrected the wording., thank you. L 117-118

13. Line 118ff.:...under the RCP8.5 scenario...

From my point of view, it would be good to briefly draw a reference to the other scenarios in order to be able to better classify the results. For example, by pointing out that the approach used shows the greatest possible impact, whereas small effects are to be expected when other RCPs are used.

Thank you for pointing out. As of the other Reviewers' comments, we have substantially modified the manuscript and included additional scenarios in our analyses.

14. Line 141ff.:...in 2100...

What does this time indication mean? Since it is a 30-year average, different time periods are possible, such as 2071-2100 or similar. Please specify exactly.

As we elaborate in the text, we compare the relative change between the simulation start (2014) and the end (2100) (L 148f.). We rephrased the section to clarify what was done. See also Lines 544ff.

15. Line 142ff.:...the simulation (2014)....

Why was 2014 chosen as the start of the simulation? Is this for technical or other practical reasons?

Thank you for asking. This was for data availability reasons. We have clarified this aspect in the text (L 551)

Line 151 and others:.....significant trend...

How was significant defined? When is a trend called significant? Since there are different approaches for testing the significance of data, further information would be useful.

We examined each quantity development using Mann-Kendall linear trend test and derived the relative development in percent from a linear fit using Theil-Sen slope. We considered a trend

significant for $p < 0.05$. We elaborate this aspect in L 559. Further statements on the significance can be found in Lines 160, 228, Figures 1d, 4d and caption of Figure 2.

16. Line 216:...(2070-2100)...
I think it should be 2071-2100.

Thank you. We have corrected this (L 272).

17. Line 243:... We do not findincreasing mean trends...
How does this fit with the statement that the amount of precipitation increases in the year?

As pointed out in the introduction (L 39ff.): [...] analyses based on climate projections show opposing trends in terms of water availability, with a slight increase in annual precipitation sums, i.e. more water, but at the same time a significant temperature increase of several degrees Celsius by 2100, i.e. less water. The resulting effect on groundwater resources is therefore not directly clear and needs to be analyzed” This is the motivation of our study, to find the future GWL trends despite intuitively opposing trends in the groundwater level forcings. Moreover, besides some regional/local differences, especially regarding future precipitation trends, which of the forcings (T or P) is the dominant one also depends on the individual site. We hope this clarifies your question.

18. Line 272:... Even fewer significant shift...
Are there any classification steps for significance?

We apologize, this is a misunderstanding. We did not mean less significant but less frequently significant. However, this section is not part of the manuscript anymore.

19. Line 284 and ff:... that temperature is mainly the driving factor for declining groundwater levels...
It should be explicitly mentioned here that the results only apply to shallow aquifers. It would also make sense to define what is meant by “shallow aquifer”. Finally, it could also be helpful to address the issue of the behavior of different aquifer types in the discussion.

Thank you for pointing out. We have added this to the respective sentence (L.322). However, we do not think that the small number of fractured and karstic aquifers (especially considering the already existing sources of uncertainty) allow a discussion of behavior differences compared to porous aquifers.

Reviewers' Comments:

Reviewer #1:

Remarks to the Author:

Please see the pdf file.

Review of revised manuscript, Deep learning shows declining groundwater levels in Germany until 2100 due to climate change

This study focused on predicting future groundwater levels in Germany using a 1D convolutional neural network (CNN) model. While the study is interesting and touches up future climates, I found its scope is narrow and provides little additional insight to the Nat Comm readers. In particular, I have the following comments.

- Groundwater aquifers are an integral component of the global terrestrial water cycle. Many studies, especially those from the GRACE community (e.g., Rodell et al. 2019; Figure 1 pasted below), have already shown a detectable downward terrestrial water storage (TWS) trend in the Germany/Austria region in recent decades. Further, a recent study conducted by Pokhrel et al. (2021, Figure 1 pasted below) demonstrated the future TWS drying trend for Europe under RCP2.6 and 6.0 by using a large ensemble of global hydrological models. Here the authors only considered shallow unconfined aquifers in a temperate climate. The data-driven projections naturally follow the same trends manifested in the climate forcings (i.e., precipitation and temperature) that the authors used. In other words, putting aside human interventions which the authors didn't consider, the results mainly reflect the causal relationship between the climate forcing and shallow groundwater storage. This is hydrology 101. However, focusing on a small region using a relatively small dataset and a purely data-driven ML approach also puts the scope of this study less significant for a high-impact journal like Nat. Comm. It'll be easier for me to recommend the publication of this article on HESS or JoH.
- Regarding LSTM, I read the HESS algorithm comparison paper published by the same authors in April [Wunsch, A., Liesch, T. & Broda, S. Groundwater level forecasting with artificial neural networks: a comparison of long short-term memory (LSTM), convolutional neural networks (CNNs), and non-linear autoregressive networks with exogenous input (NARX). *Hydrology Earth System Sciences* 25, 1671–1687 (2021)]. There the authors considered an autoregressive setup, $GW_t = f(GW_{t-1}, \dots, t-N)$, namely, they incorporated antecedent GW to predict future GW. It is known (e.g., Selvin et al., 2017) using LSTM in an autoregressive setting may hurt its performance due to noise in data. There are ways to alleviate that effect (e.g., using moving average Feng et al., 2020). Under this work, however, the authors mainly used precip and temperature data to drive the ML model. Using LSTM in the setting of this work has been shown to achieve state-of-the-art performance (Kratzert et al., 2019) compared to physics-based models.
- I think the power of ML has been underutilized in this work by doing single well predictions. Existing works have already shown the merits of incorporating a large sample dataset to perform many-to-one prediction, which can be especially important given the levels in many wells can be spatially correlated. Existing works have also compared the performance of ML to similar process-based models, or at least adopting a physics-informed ML approach. This work does not possess those elements.
- One of the main selling points of this work is using a data-driven model to project the future scenarios. However, numerous climate studies already pointed the caveat of this approach in capturing future extremes (<https://phys.org/news/2018-07-machine-method-capable-accurate-extrapolation.html>). Although the time dimension was extrapolated, the extremes of predicted groundwater levels cannot be extrapolated if they are not part of the historical data used for

training. A true extrapolation would instead learn the distribution of data (i.e., generative modeling), from which the tails of distribution can be extrapolated. That's not the approach taken by the authors. In their rebuttal letter, the authors mentioned "The changes in future climate patterns we see (e.g. increasing temperatures) are changes in mean values and absolute future values usually still are in the range of the training data" This is a very irresponsible subjective statement. How can you see the future without validation? If the future patterns are truly like the current days as you mentioned, then what is the point of projection. You can simply apply the groundwater climatology.

Fig. 1 | Annotated map of TWS trends. Trends in TWS (in centimetres per year) obtained on the basis of GRACE observations from April 2002 to March 2016. The cause of the trend in each outlined study region is briefly explained and colour-coded by category. The trend map was smoothed

with a 150-km-radius Gaussian filter for the purpose of visualization; however, all calculations were performed at the native 3° resolution of the data product.

Fig. 1 | Impact of climate change on TWS. **a-d**, The changes (multi-model weighted mean) in TWS, averaged for the mid- (2030–2059; **a,c**) and the late (2070–2099; **b,d**) twenty-first century under RCP2.6 (**a,b**) and RCP6.0 (**c,d**) relative to the average for the historical baseline period (1976–2005). The colour hues show the magnitude of change and the saturation indicates the agreement, among ensemble members, in the sign of change. The graph on the right of each panel shows the latitudinal mean.

References:

- Rodell, M., Famiglietti, J. S., Wiese, D. N., Reager, J. T., Beaudoin, H. K., Landerer, F. W., & Lo, M. H. (2019). Emerging trends in global freshwater availability (vol 557, pg 651, 2018). *Nature*, 565(7739), E7-E7.
- Pokhrel, Y., Felfelani, F., Satoh, Y., Boulange, J., Burek, P., Gädeke, A., ... & Wada, Y. (2021). Global terrestrial water storage and drought severity under climate change. *Nature Climate Change*, 11(3), 226-233.
- Selvin, S., Vinayakumar, R., Gopalakrishnan, E. A., Menon, V. K., & Soman, K. P. (2017, September). Stock price prediction using LSTM, RNN and CNN-sliding window model. In 2017 international conference on advances in computing, communications and informatics (icacci) (pp. 1643-1647). IEEE.
- Feng, D., Fang, K., & Shen, C. (2020). Enhancing streamflow forecast and extracting insights using long-short term memory networks with data integration at continental scales. *Water Resources Research*, 56(9), e2019WR026793.
- Kratzert, F., Klotz, D., Herrnegger, M., Sampson, A. K., Hochreiter, S., & Nearing, G. S. (2019). Toward improved predictions in ungauged basins: Exploiting the power of machine learning. *Water Resources Research*, 55(12), 11344-11354.

Reviewer #2:

Remarks to the Author:

The revisions of Wunsch et al. clearly improved the manuscript which benefits from adding RCP 2.6 & 4.5 as well as additional uncertainty analysis (Theil-Sen line) and discussion (ll. 347-367). However, from my perspective there remain two points for further revision:

- The authors use the MK-trend test without pre-whiting. It is important to note that this test is only valid in case of no autocorrelation, otherwise the significance of the test will be overestimated. The authors correctly state that the seasonal autocorrelation is not relevant in the context of their MK-test as they only use annual values. However, apart from seasonal autocorrelation groundwater often exhibits autocorrelation on longer time scales as well. Based on my experience with groundwater data in Germany I would expect at least half of the groundwater records to show significant first-order autocorrelation for the annual values used in this work.

As an example, I made a quick test and randomly selected one of the wells used for this work (file "NI_40000175_GW-Data.csv" from the repository). After calculating annual time series of the mean and 2.5- and 97.5-percentiles I get first-order autocorrelations of 0.35 (mean), 0.02 (2.5-percentile) and 0.65 (97.5-percentile). The correlations for the mean and the 97.5.-percentile are significant and definitely relevant in the context of MK-trend tests. If the author's CNN model captures the dynamics of the well correctly, the modelled time series until 2100 will exhibit a similar autocorrelation structure and without appropriate pre-processing the MK-test will overestimate trend significance for this well. Hence, the authors will definitely have to check for autocorrelation and exclude where existent before using MK and characterizing trends as significant.

- The authors discuss different sources of uncertainty including model uncertainty. However, models are usually trained to match mean conditions best but are much weaker in simulating extremes. Based on Figure 7 this seems to be the case also for the models used in this work as it can be clearly seen that the maxima of the time series are systematically underestimated. The plots in the supplements reveal similar problems for the extremes (mostly the upper extreme, sometimes also the lower extreme) at many stations. Hence, the studies' results regarding the mean will be more robust than those regarding the extremes. To judge the validity of model results I think it is necessary to evaluate the model performance regarding the different metrics (annual mean, annual 2.5-/97.5-percentile) in more detail.

Reviewer #3:

Remarks to the Author:

Comments:

By revising the paper, the statements were once again clearly sharpened.

It is now clear which general statements about groundwater development in Germany are possible and where the results still showed larger bandwidths that do currently not allow clear statements. With regard to climate impacts, it was clearly shown what influence global warming can have on groundwater availability in Germany. Furthermore, the results also clearly show that any global reduction in CO₂-emissions will have a positive impact on groundwater level and groundwater yield. However, the results also show that the resource groundwater will change regionally in the future and that all users must adapt to this.

From my point of view, I only have two small comments:

Line 41 ff.: this sentence is formulated somewhat unclearly. Actually, all models show a robust increase in temperature (i.e. (almost) all climate models agree on this), but there are drier and wetter models for precipitation, depending on the calculation approach. However, these statements cannot be read out of the text clearly.

Line 56: to meet the needs.... Wouldn't it also be important to mention here that climate change, especially higher temperatures, also has an impact on changing water demands (not only in the city).

This is particularly relevant when considering peak demands. This addition is not an absolute must, but could build an important bridge to practice.

Response to Reviewers

We thank all Reviewers for the repeated revision of our manuscript. We are glad to read that our revisions from stage one sharpened the results. We will comment on the open questions and concerns in the following. Please find the reviewers comments in black and our answer in red. Line numbers in our answers refer to the newly revised manuscript.

Dear Mr. Wunsch,

Thank you again for submitting your manuscript "Deep learning shows declining groundwater levels in Germany until 2100 due to climate change" to Nature Communications. We have now received reports from 3 reviewers and, on the basis of their comments, we have decided to invite a revision of your work for further consideration in our journal. Your revision should address all the points raised by our reviewers (see their reports below). In particular, Reviewer #1 and #2 raise important technical concerns, such as the presence of autocorrelation and underutilization of the machine learning methods that must be addressed for publication in Nature Communications.

When resubmitting, you must provide a point-by-point response to the reviewers' comments. Please show all changes in the manuscript text file with track changes or colour highlighting. If you are unable to address specific reviewer requests or find any points invalid, please explain why in the point-by-point response.

REVIEWER COMMENTS

Reviewer #1 (Remarks to the Author):

Review of revised manuscript, Deep learning shows declining groundwater levels in Germany until 2100 due to climate change

This study focused on predicting future groundwater levels in Germany using a 1D convolutional neural network (CNN) model. While the study is interesting and touches up future climates, I found its scope is narrow and provides little additional insight to the Nat Comm readers. In particular, I have the following comments.

We thank anonymous Reviewer#1 for again reviewing our manuscript. We recognize that no recommendation for publication Nature Communications is provided. With surprise, we found that in the second review stage some new fundamental criticism is raised that (i) has not been mentioned in stage one and (ii) that no constructive comments or propositions are given other than changing the fundamental approach including both data basis and methods. We therefore were not able to change our manuscript accordingly, however, we try to comment on every point in the following.

1. Groundwater aquifers are an integral component of the global terrestrial water cycle. Many studies, especially those from the GRACE community (e.g., Rodell et al. 2019; Figure 1 pasted below), have already shown a detectable downward terrestrial water storage (TWS) trend in the Germany/Austria region in recent decades. Further, a recent study conducted by Pokhrel et al. (2021, Figure 1 pasted below) demonstrated the future TWS drying trend for Europe under RCP2.6 and 6.0 by using a large ensemble of global hydrological models. Here the authors only considered shallow unconfined

aquifers in a temperate climate. The data-driven projections naturally follow the same trends manifested in the climate forcings (i.e., precipitation and temperature) that the authors used. In other words, putting aside human interventions which the authors didn't consider, the results mainly reflect the causal relationship between the climate forcing and shallow groundwater storage. This is hydrology 101. However, focusing on a small region using a relatively small dataset and a purely data-driven ML approach also puts the scope of this study less significant for a high-impact journal like Nat. Comm. It'll be easier for me to recommend the publication of this article on HESS or JoH.

We are pleased that Reviewer #1 recognizes that our models “reflect the causal relationship between the climate forcing and shallow groundwater” as it was one of our main goals during model building and training to ensure that the deep learning models learn the correct relationship. Groundwater dynamics and groundwater recharge seem simple, yet they are complex processes, depending on many boundary conditions and controlling factors, that superimpose each other in time and space (otherwise there would be no need for complex groundwater models). It is therefore not as simple as implied by the statements above, to derive groundwater levels from climate forcings alone. Much more important, we show that precipitation and temperature (depending on the scenario) regionally influence groundwater in possibly contradictory ways (e.g. L. 41-43, and L. 116ff). It is therefore not obvious from the input data alone which direction the future groundwater level development will follow. We further show that the calculated trends and changes in our study indeed do not simply reflect the spatial input data patterns (e.g. for RCP8.5, as mentioned in L. 208-209).

In comparison to global studies, we base our analyses on specifically suitable climate projections, namely the “core-ensemble” of the German Meteorological Service. All members of this ensemble fulfill certain quality and validity criteria for central Europe. We therefore think that our study nicely complements existing studies (such as Pokhrel et al. (2021)) by investigating regional climate change effects with (slightly) reduced input data uncertainty. Especially compared to the mentioned study of Pokhrel et al. (2021), we provide additional insights by investigating three instead of two RCP scenarios and by including more than four (RCP2.6: five, 4.5 and 8.5: six) climate models for each scenario. Thus, we potentially better represent the full range of possible developments across different RCP paths as well within each scenario. In comparison to results focusing on TWS in general, simulation of groundwater levels does not only reveal a reduction in total water availability but allows conclusions on the future variability of the important water resource of groundwater and specific effects on wet and dry periods. Moreover, it is well-known that GRACE derived data (e.g. Rodell et al. 2019) has a much coarser resolution and is therefore not suitable for regional or even local studies.

2. Regarding LSTM, I read the HESS algorithm comparison paper published by the same authors in April [Wunsch, A., Liesch, T. & Broda, S. Groundwater level forecasting with artificial neural networks: a comparison of long short-term memory (LSTM), convolutional neural networks (CNNs), and non-linear autoregressive networks with exogenous input (NARX). Hydrology Earth System Sciences 25, 1671–1687 (2021)]. There the authors considered an autoregressive setup, $GW_t = f(GW_{\{t-1, \dots, t-N\}})$, namely, they incorporated antecedent GW to predict future GW. It is known (e.g., Selvin et al., 2017) using LSTM in an autoregressive setting may hurt its performance due to noise in data. There are ways to alleviate that effect (e.g., using moving average Feng et al., 2020). Under this work, however, the authors mainly used precip and temperature data

to drive the ML model. Using LSTM in the setting of this work has been shown to achieve state-of-the-art performance (Kratzert et al., 2019) compared to physics-based models.

We thank Reviewer#1 for reading our HESS study, but we also think there exists a misunderstanding. We therefore would like to point out that while we have investigated an autoregressive setup in the above-mentioned study in the context of short-term forecasting without input data, this setup is of no relevance for the submitted manuscript. The larger part of the HESS study investigated sequence-to-value or sequence-to-one prediction solely based on meteorological input forcings (precipitation, temperature and relative humidity in this case), thus similar to the approach chosen in this manuscript and also comparable to the approach by Kratzert et al. (2019). This is what we refer to and what we base our conclusions on, regarding the appropriateness of CNN models.

3. I think the power of ML has been underutilized in this work by doing single well predictions. Existing works have already shown the merits of incorporating a large sample dataset to perform many-to-one prediction, which can be especially important given the levels in many wells can be spatially correlated. Existing works have also compared the performance of ML to similar process-based models, or at least adopting a physics-informed ML approach. This work does not possess those elements.

We agree that there is plenty of room for improvement in future studies and that our methodology is not the be-all and end-all. However, in our opinion we also demonstrated sufficiently that our results are valid and allow reasonable conclusions on the future groundwater level development in Germany.

It is true that performing many-to-one predictions outperformed existing models in rainfall-runoff modeling (Kratzert et al. 2019), however, we lack comparable data to transfer this approach to our study area and the groundwater domain. Theoretically, therefore, the power of ML was underutilized, but practically it was not. For future studies we are already working on the collection of such data, which, however, is not yet available. Generally, it remains to say that we see advantages of many-to-one approaches not in spatial correlation, but in improved extrapolation capabilities of a model. Especially in the context of climate change, knowledge transfer from locations with historically different conditions can help to estimate the reaction to previously unseen climate at a site. To fully exploit this advantage, one should even include additional regions, other than Germany, to enable the model to learn different climate conditions historically.

Regarding “physics-informed”, we would argue that this phrase itself is not yet well defined. Many studies sell simple model modifications as physics-informed (e.g. not allowing below-zero output values if physically not reasonable, or even only “physics-informed” input features), while only few studies incorporate true physics (such as mass conservation restraints) in their models. One could even argue that our models are at least physics-controlled, as we used XAI to check the conceptual correctness of our models. We agree, however, that there is great potential to improve simulations using physics in models for the future.

4. One of the main selling points of this work is using a data-driven model to project the future scenarios. However, numerous climate studies already pointed the caveat of this approach in capturing future extremes (<https://phys.org/news/2018-07-machine-method-capable-accurate-extrapolation.html>). Although the time dimension was extrapolated, the extremes of predicted groundwater levels cannot be extrapolated if they are not part of the historical data used for training. A true extrapolation would

instead learn the distribution of data (i.e., generative modeling), from which the tails of distribution can be extrapolated. That's not the approach taken by the authors. In their rebuttal letter, the authors mentioned "The changes in future climate patterns we see (e.g. increasing temperatures) are changes in mean values and absolute future values usually still are in the range of the training data" This is a very irresponsible subjective statement. How can you see the future without validation? If the future patterns are truly like the current days as you mentioned, then what is the point of projection. You can simply apply the groundwater climatology.

We admit that we have used inaccurate wording in our last response letter, as we spoke of "patterns", which of course are not similar in the future and the past. However, we formulated our manuscript with more caution and would like to refer to the respective sentence there: "[...] because mean values and frequencies of input values change in the future, but the total range of these values is usually already present in the training data." As we speak of "range" instead of "patterns", this formulation is more precise and better represents what we tried to express.

Fig. 1 | Annotated map of TWS trends. Trends in TWS (in centimetres per year) obtained on the basis of GRACE observations from April 2002 to March 2016. The cause of the trend in each outlined study region is briefly explained and colour-coded by category. The trend map was smoothed

with a 150-km-radius Gaussian filter for the purpose of visualization; however, all calculations were performed at the native 3° resolution of the data product.

Fig. 1 | Impact of climate change on TWS. a-d, The changes (multi-model weighted mean) in TWS, averaged for the mid- (2030–2059; **a,c**) and the late (2070–2099; **b,d**) twenty-first century under RCP2.6 (**a,b**) and RCP6.0 (**c,d**) relative to the average for the historical baseline period (1976–2005). The colour hues show the magnitude of change and the saturation indicates the agreement, among ensemble members, in the sign of change. The graph on the right of each panel shows the latitudinal mean.

References:

- Rodell, M., Famiglietti, J. S., Wiese, D. N., Reager, J. T., Beaudoin, H. K., Landerer, F. W., & Lo, M. H. (2019). Emerging trends in global freshwater availability (vol 557, pg 651, 2018). *Nature*, 565(7739), E7E7.
- Pokhrel, Y., Felfelani, F., Satoh, Y., Boulange, J., Burek, P., Gädeke, A., ... & Wada, Y. (2021). Global terrestrial water storage and drought severity under climate change. *Nature Climate Change*, 11(3), 226233.
- Selvin, S., Vinayakumar, R., Gopalakrishnan, E. A., Menon, V. K., & Soman, K. P. (2017, September). Stock price prediction using LSTM, RNN and CNN-sliding window model. In 2017 international conference on advances in computing, communications and informatics (icacci) (pp. 1643-1647). IEEE.
- Feng, D., Fang, K., & Shen, C. (2020). Enhancing streamflow forecast and extracting insights using longshort term memory networks with data integration at continental scales. *Water Resources Research*, 56(9), e2019WR026793.
- Kratzert, F., Klotz, D., Herrnegger, M., Sampson, A. K., Hochreiter, S., & Nearing, G. S. (2019). Toward improved predictions in ungauged basins: Exploiting the power of machine learning. *Water Resources Research*, 55(12), 11344-11354.

Reviewer #2 (Remarks to the Author):

The revisions of Wunsch et al. clearly improved the manuscript which benefits from adding RCP 2.6 & 4.5 as well as additional uncertainty analysis (Theil-Sen line) and discussion (ll. 347-367). However, from my perspective there remain two points for further revision:

- The authors use the MK-trend test without pre-whiting. It is important to note that this test is only valid in case of no autocorrelation, otherwise the significance of the test will be overestimated. The authors correctly state that the seasonal autocorrelation is not relevant

in the context of their MK-test as they only use annual values. However, apart from seasonal autocorrelation groundwater often exhibits autocorrelation on longer time scales as well. Based on my experience with groundwater data in Germany I would expect at least half of the groundwater records to show significant first-order autocorrelation for the annual values used in this work.

As an example, I made a quick test and randomly selected one of the wells used for this work (file "NI_40000175_GW-Data.csv" from the repository). After calculating annual time series of the mean and 2.5- and 97.5-percentiles I get first-order autocorrelations of 0.35 (mean), 0.02 (2.5-percentile) and 0.65 (97.5-percentile). The correlations for the mean and the 97.5.-percentile are significant and definitely relevant in the context of MK-trend tests. If the author's CNN model captures the dynamics of the well correctly, the modelled time series until 2100 will exhibit a similar autocorrelation structure and without appropriate pre-processing the MK-test will overestimate trend significance for this well.

Hence, the authors will definitely have to check for autocorrelation and exclude where existent before using MK and characterizing trends as significant.

Thank you for raising this important concern. We checked our calculations and first-order autocorrelation had indeed a certain influence on the results presented in our manuscript. We therefore applied 3PW method (mannkendall/Python package) after Collaud Coen et al. (2020). Advantage of this method is the combination of three pre-whitening approaches to overcome shortcomings and assumptions of each approach. Overall, slightly fewer results are considered significant now and some changes are considered a bit weaker; however, the general conclusions of our analyses still remain. We adapted the newly calculated trends to all relevant graphics in the manuscript and the supplement and corrected the text accordingly.

References:

Collaud Coen, M. *et al.* Effects of the prewhitening method, the time granularity, and the time segmentation on the Mann–Kendall trend detection and the associated Sen's slope. *Atmos. Meas. Tech.* **13**, 6945–6964 (2020).

Vogt, F. P. A. mannkendall/Python. (Zenodo, 2021). doi:10.5281/ZENODO.4495590.

- The authors discuss different sources of uncertainty including model uncertainty. However, models are usually trained to match mean conditions best but are much weaker in simulating extremes. Based on Figure 7 this seems to be the case also for the models used in this work as it can be clearly seen that the maxima of the time series are systematically underestimated. The plots in the supplements reveal similar problems for the extremes (mostly the upper extreme, sometimes also the lower extreme) at many stations. Hence, the studies' results regarding the mean will be more robust than those regarding the extremes. To judge the validity of model results I think it is necessary to evaluate the model performance regarding the different metrics (annual mean, annual 2.5-/97.5-percentile) in more detail.

Thank you for pointing out, this indeed deserves a detailed discussion. With validity, we can judge the model performance only for the comparably short testing period of 4 years, which makes it especially difficult to derive conclusions for extreme value performance, as mostly only four highs and four lows occur in four years. Nevertheless, we evaluated the relative model Bias (normalized on the historic Min-Max range), for this period and for all models. We included a discussion of this evaluation in the uncertainty section of our manuscript. We hope that this sharpens the different

sources of uncertainty for the reader. We agree that in general the estimation of the mean conditions in the future is more robust than for the extreme values. However, we also think that due to the systematic nature of this error (even though difficult to quantify), that relative trends or tendencies derived from these models, still are reasonably interpretable, even for the extreme values.

Reviewer #3 (Remarks to the Author):

Comments:

By revising the paper, the statements were once again clearly sharpened. It is now clear which general statements about groundwater development in Germany are possible and where the results still showed larger bandwidths that do currently not allow clear statements. With regard to climate impacts, it was clearly shown what influence global warming can have on groundwater availability in Germany. Furthermore, the results also clearly show that any global reduction in CO₂-emissions will have a positive impact on groundwater level and groundwater yield. However, the results also show that the resource groundwater will change regionally in the future and that all users must adapt to this.

From my point of view, I only have two small comments:

- Line 41 ff.: this sentence is formulated somewhat unclearly. Actually, all models show a robust increase in temperature (i.e. (almost) all climate models agree on this), but there are drier and wetter models for precipitation, depending on the calculation approach. However, these statements cannot be read out of the text clearly.

Thank you for pointing out. This is indeed an important aspect that should become clear from the text. We have modified this statement to be more precise. L 39-42

- Line 56: to meet the needs.... Wouldn't it also be important to mention here that climate change, especially higher temperatures, also has an impact on changing water demands (not only in the city). This is particularly relevant when considering peak demands. This addition is not an absolute must, but could build an important bridge to practice.

Thank you for pointing out this important aspect. We agree that water demands not only increase in urban areas but also in rural areas for example due to agricultural irrigation. In this sentence we already list growing population/urban areas, industry and agricultural irrigation. It seems that it nevertheless has not become clear that we refer to all of these factors, therefore we now slightly modified the wording. L 56-60

Response to Reviewers

Please find our statements (red) to the reviewers' comments (black) in the following.

Reviewer #2 (Remarks to the Author):

I think the authors have replied satisfactorily to all points raised by the referees. The changes have further improved the quality of the paper and I have no other concerns.

Thank you for your constructive criticism during the review process, which helped to improve the manuscript substantially. We are glad to read that there are no more concerns.